# HIV-associated gut dysbiosis is independent of sexual practice and correlates with noncommunicable diseases

I. Vujkovic-Cvijin [1,11], O. Sortino [2,3,11], E. Verheij [4], J. Sklar [1], F. W. Wit [4], N. A. Kootstra [5], B. Sellers [6], J. M. Brenchley [7], J. Ananworanich [4,8,9], M. Schim van der Loeff [10], Y. Belkaid [1,6], P. Reiss [4] & I. Sereti [3✉]

Loss of gut mucosal integrity and an aberrant gut microbiota are proposed mechanisms contributing to chronic inflammation and increased morbidity and mortality during antiretroviral-treated HIV disease. Sexual practice has recently been uncovered as a major source of microbiota variation, potentially confounding prior observations of gut microbiota alterations among persons with HIV (PWH). To overcome this and other confounding factors, we examine a well-powered subset of AGEhIV Cohort participants comprising antiretroviral-treated PWH and seronegative controls matched for age, body-mass index, sex, and sexual practice. We report significant gut microbiota differences in PWH regardless of sex and sexual practice including *Gammaproteobacteria* enrichment, *Lachnospiraceae* and *Ruminococcaceae* depletion, and decreased alpha diversity. Men who have sex with men (MSM) exhibit a distinct microbiota signature characterized by *Prevotella* enrichment and increased alpha diversity, which is linked with receptive anal intercourse in both males and females. Finally, the HIV-associated microbiota signature correlates with inflammatory markers including suPAR, nadir CD4 count, and prevalence of age-associated noncommunicable comorbidities.

[1] Metaorganism Immunity Section, Laboratory of Immune System Biology, National Institute of Allergy and Infectious Diseases (NIAID), National Institutes of Health (NIH), Bethesda, MD 20892, USA. [2] Clinical Monitoring Research Program Directorate, Frederick National Laboratory for Cancer Research, National Cancer Institute, Bethesda, USA. [3] HIV Pathogenesis Section, Laboratory of Immunoregulation, NIAID, NIH, Bethesda, MD 20892, USA. [4] Amsterdam University Medical Centers, University of Amsterdam, Department of Global Health and Division of Infectious Diseases, Amsterdam Infection and Immunity Institute, Amsterdam Public Health Research Institute, and Amsterdam Institute for Global Health and Development, Amsterdam, Netherlands. [5] Amsterdam University Medical Centers, University of Amsterdam, Department of Experimental Immunology, Amsterdam Infection & Immunity Institute, Amsterdam, Netherlands. [6] Trans-NIH Center for Human Immunology, Autoimmunity, and Inflammation, National Institutes of Health, Bethesda, MD 20892, USA. [7] Barrier Immunity Section, Laboratory of Viral Diseases, NIAID, NIH, Bethesda, MD 20892, USA. [8] SEARCH/Thai Red Cross AIDS Research Centre, Bangkok, Thailand. [9] United States Military HIV Research Program, Walter Reed Army Institute of Research, Silver Spring, MD 20910, USA. [10] Department of Infectious Diseases, Public Health Service of Amsterdam, Amsterdam, Netherlands. [11]These authors contributed equally: I. Vujkovic-Cvijin, O. Sortino. A list of the AGEhIV Cohort Study group members and affiliations is given in Supplementary Note 1. ✉email: isereti@niaid.nih.gov

Effective antiretroviral therapy (ART) has prolonged survival and shifted the morbidity spectrum for persons with HIV (PWH) from AIDS toward age-associated noncommunicable comorbidities including cardiovascular, osteoporotic, metabolic, hepatic, and renal disease, conditions that occur with increased incidence compared to age-matched HIV-uninfected individuals[1]. A key contributor to the current disease spectrum is HIV-associated inflammation and immune activation that persists in chronically treated HIV infection[2], the etiology of which remains incompletely defined. Mounting evidence supports the concept that the gut microbiota has a vital role in maintaining immune homeostasis and can, when its composition becomes aberrant, spur pathological, chronic inflammation in a variety of conditions[3]. Numerous studies have suggested that the gut microbiota contributes to HIV-associated inflammation, though illumination of its role in this process is hampered by an incomplete understanding of the precise identities of microbiota members that are altered in PWH.

The microbiota of the gastrointestinal tract is a tightly regulated community of microorganisms that has dramatic effects on host physiology and immune processes. For example, *Akkermansia muciniphila* can promote intestinal mucosal homeostasis through modulation of mucus thickness[4]. Abundance of certain *Lachnospiraceae* and *Ruminococcaceae* species in the gut induces expansion of regulatory T cells that can down-regulate harmful inflammatory responses[5]. Furthermore, short-chain fatty acid (SCFA) metabolites, derived from commensal bacteria in these clades and others, promote intestinal barrier integrity via their role in epithelial cell energy metabolism and induction of regulatory T cells[6]. Derangement of abundances of these microbes can thus impact the aforementioned protective physiological phenomena. Furthermore, enrichment of invasive, translocative *Gammaproteobacteria* in mice can be sufficient to cause chronic systemic inflammation[7]. Several components of the gut immune barrier that are responsible for regulating composition of the gut microbiota are abnormal in HIV infection, including Paneth cells[8], macrophages[9], epithelial cells[10,11], and $T_H17$ cells[12], giving rise to the possibility of an altered microbiota that results from HIV infection. In addition, these gastrointestinal barrier components are important for preventing translocation of microbial products into the circulation, a process that has been postulated to contribute to the chronic inflammation that spurs age-associated noncommunicable comorbidities in PWH. Indeed, in vitro studies have shown a higher pro-inflammatory capacity of gut bacteria from treated PWH[13] compared to seronegative participants, and prophylactic antibiotic treatment that alters the microbiota also reduces levels of intestinal inflammation[14]. Importantly, our understanding of which gut bacteria are responsible for these phenotypes, and whether homeostasis-promoting or pro-inflammatory gut bacteria are differentially abundant in PWH, remains incomplete.

Studies of the gut microbiota in HIV infection exhibit great heterogeneity with regard to the characteristics of participants investigated, cohort sizes, type of sampling method used, sequencing depth, and level of bacterial taxonomic classifications reported. Despite this, some common features in microbiota composition of PWH have been reported[15–24]. These include an over-representation of *Gammaproteobacteria* within the *Enterobacteriaceae* and *Desulfovibrionaceae* families, of which several are known as having pro-inflammatory properties and have been linked to plasma levels of innate immune activation markers in PWH[15,16]. In addition, a decline in the frequencies of *Lachnospiraceae*, *Ruminococcaceae*, *Rikenellaceae* and *Bacteroides*—bacterial taxa linked with anti-inflammatory properties and maintenance of gut homeostasis—have been reported in PWH with varying degrees of consistency[15–24]. Studies have also

described an increased abundance of *Prevotella* in PWH as compared to HIV-uninfected controls[18,20–22], though these taxonomic shifts have been observed with less consistency.

Recent seminal studies have reported an increased abundance of *Prevotella* and depletion of *Bacteroides* in the gut microbiota among men who have sex with men (MSM) as compared to men who have sex with women (MSW), independently of HIV-1 infection status[24–26]. Because the HIV-infected population in many resource-rich settings is predominantly comprised of MSM, selection of seronegative controls from the general population without matching for MSM could therefore have confounded prior studies examining the impact of HIV infection on the gut microbiota. Studies for which an uninfected control population was not deliberately selected from the MSM population thus likely measured the impact on the gut microbiota of both being MSM and HIV-infected, as opposed to singly examining microbiota associations with HIV infection. These important revelations have raised the hypothesis that after stratification by sexual practice, there may be little to no evidence for shifts in the gut microbiota in PWH compared to seronegative controls, and thus that prior reports of HIV-associated shifts in the gut microbiota were in actuality capturing solely MSM-associated microbiota shifts. Alternately, MSM status and HIV infection status may both be microbiota-modulating factors that each exert unique, orthogonal effects on the microbiota, in which case it remains to be determined which taxonomic shifts described in prior reports were driven by which microbiota-modulating factor (i.e. PWH or MSM status).

In light of these recent findings and resulting competing hypotheses, a lack of well-powered studies in which PWH and controls are matched for variables that may confound microbiota analyses has limited our understanding of the role of the gut microbiota in HIV disease. Variables previously reported to impact the microbiota—and thus to confound comparisons if cases and controls are not matched for these variables—include sex, age, sexual practice, body mass index (BMI), and immigration status, and have not been tracked in well-powered, evenly represented cohorts. To achieve a high resolution analysis of the impact of HIV infection on the gut microbiota, a large cohort is necessary in order to accommodate selection of a population that is heterogeneous enough in the aforementioned variables to appropriately represent the human population while also being carefully matched for these variables between PWH cases and seronegative controls. In the present study, we investigate the fecal microbiota profiles of a well-powered cohort of chronically HIV-infected people with suppressed viremia on antiretroviral therapy (ART) and HIV-uninfected controls, stratified by sex (i.e. male and female) and sexual practice (i.e. lifetime or recent vaginal/anal intercourse with women/men) and matched for age, sexual practice, BMI, and birth country. We present evidence for distinct and opposing microbiota signatures of both MSM status and HIV infection status, and in addition present links between gut microbiota features in HIV infection and both inflammatory markers and prevalence of noncommunicable comorbidities in this population.

## Results
**Clinical characteristics and demographics of participants.** A cohort of matched PWH and HIV-seronegative controls (total $n = 160$, PWH $n = 80$, HIV-seronegative controls $n = 80$), was assembled to include stratified subgroups of 83 MSM, 38 non-MSM males (MSW, defined as male participants that reported never having sex with men) and 39 females. When examining each subgroup individually, the female group [F = HIV-infected (PWH-F) and HIV-seronegative controls (SN-F)]; the men who

have sex with men group [MSM = HIV-infected (PWH-MSM) and HIV-seronegative controls (SN-MSM)] and the non-MSM male group [MSW = HIV-infected (PWH-MSW) and HIV-seronegative controls (SN-MSW)], there were no significant differences in age, BMI, tobacco smoking status, receptive anal intercourse within the past 6 months (RAI), lifetime partners, alcohol intake frequency, or amphetamine use among the PWH and seronegative controls (Supplementary Data 1). All participants were residents of the Netherlands and although some differed in their birth countries, birth countries did not differ significantly between PWH and seronegative controls within each subgroup (Supplementary Data 1). Ninety-one percent of the cohort completed a questionnaire regarding sexual practice during the 6 month period preceding their study visit. Sixteen percent of PWH-F, and ten percent of SN-F declared to have had receptive anal intercourse within the past six months (RAI) ($P = 0.298$ Fisher's exact test). Forty-eight percent of PWH-MSM and sixty-two percent of SN-MSM engaged in RAI ($P = 0.334$ Fisher's exact test, Supplementary Data 1). There were no significant differences in virologic and immunologic characteristics of PWH across subject groups (Supplementary Data 2).

**Gut microbiota differs between PWH and HIV-seronegative controls**. Fecal samples were used for profiling of microbial communities via 16 rRNA sequencing. After filtering for quality and sequencing depth, confounder-matched pairs of PWH and seronegative controls totaling $n = 142$ total subjects were compared (36 females, 72 MSM, 34 MSW). HIV infection status was significantly associated with a decrease in all common measures of alpha diversity (Fig. 1a), supporting a recent meta-analysis[27]. Principal coordinates analysis (Fig. 1b) revealed clustering of PWH and HIV-seronegative controls which was verified by permutational multivariate analysis of variance ($P < 0.001$, PERMANOVA). Community composition differentiated PWH from HIV-seronegative controls ($P = 0.001$, PERMANOVA) even when stratifying by group (F, MSM, MSW) and birth country, two variables that also exhibited significant impact on the gut microbiota (Supplementary Data 3). Having selected uninfected control participants that matched each PWH participant by age, sex, sexual preference, BMI, and birth country, we employed a paired-sample statistical study design to ensure comparisons were not confounded by any of the aforementioned characteristics. The paired non-parametric Wilcoxon test was thus used to identify gut bacterial taxa that differed in abundance between PWH and seronegative controls ($n = 142$ total paired samples, Supplementary Data 4). Among the bacterial taxa enriched in PWH were members of the *Desulfovibrionaceae* (specifically, *Bilophila wadworthia*) and *Enterobacteriaceae* families (Fig. 1c), whose frequency has been variably observed as increased in PWH[15–19,21–24] and in inflammatory bowel disease[28]. Taxa depleted in PWH—comprising the likely origin of the reduction in alpha diversity—were predominantly *Clostridiales* members of the *Lachnospiraceae* and *Ruminococcaceae* families, an observation also in agreement with some prior studies[16,17,20–24]. Members of these clades are known to be enriched for producers of SCFA[6], a class of compounds that provide energy for gut epithelial cells, prevent expansion of pro-inflammatory *Proteobacteria*[29], and induce differentiation of immunologic tolerance-promoting T regulatory cells[6]. These data support the conclusion that PWH exhibit an altered gut microbiota as compared to HIV-uninfected controls that may include an expansion of pro-inflammatory gut bacteria and a depletion of homeostasis-promoting microbiota members. Notable, however, was the lack of observed depletion of *Bacteroides* and enrichment of *Prevotella* taxa, which were hallmark features of some prior HIV microbiota studies[18–23].

**Microbiota differs in PWH regardless of sex/sexual practice**. To determine to what degree treated HIV infection itself and sex/sexual practice contribute to microbial community alterations in PWH, we stratified the analysis by groups: F ($n = 36$), MSM ($n = 72$) and MSW ($n = 34$). Among MSM and F, PWH exhibited decreased alpha diversity by at least one alpha diversity measure (Fig. 2a), with an equivalent trend for MSW. Shannon diversity, a measure of both species richness (the number of taxa detected in a sample) as well as evenness of their distribution, was reduced in PWH-F and exhibited the same pattern among PWH-MSM and PWH-MSW (Fig. 2a). PWH-MSM exhibited robust decreases in species richness and Faith's phylogenetic diversity (a measure of the breadth of the phylogenetic tree that is present in a sample), whereas both PWH-F and PWH-MSW also exhibited lower species richness. Overall, these data suggest a decrease in alpha diversity among PWH compared to HIV-seronegative controls regardless of sex/sexual practice. Comparisons of community structure using beta diversity analyses revealed significant differences by HIV status in each of the three groups considered individually (MSM $P = 0.005$, MSW $P = 0.018$, F $P = 0.049$; Fig. 2b). We found that amplicon sequence variants (ASVs) that were in differential abundance between all matched PWH and negative controls bore consistent abundance trends across all three groups (Fig. 2c). Furthermore, when examining the top ASVs that differentiated PWH from controls within each group considered individually, these taxa exhibited concordance in the directionality of their abundance shifts (whether enriched in PWH or controls) across the three groups (Fisher's exact test $P < 0.0001$, Supplementary Fig. 2). Together these data suggest that treated HIV infection exerts a robust effect on the gut bacterial community across human populations that pervades sex and sexual practice.

**MSM-associated microbiota is distinct from that of HIV**. Next we investigated the gut microbial community in MSM to understand the contribution of sexual practice to PWH-associated dysbiosis. Despite the fact that birth country of study participants was matched between PWH and seronegative controls within each group, subgroups (F, MSM, MSW) differed in their proportions of participant birth countries, with the predominant group being born in the country of recruitment, the Netherlands. Because immigration status is associated with microbiota differences[30], for subsequent MSM analyses we sought to mitigate confounding effects of immigration status by selecting only non-immigrant persons born in the most numerous birth country group (Netherlands). MSM exhibited significantly higher alpha diversity regardless of HIV infection status when compared to MSW and F (Fig. 3a, Supplementary Fig. 1a, b), a finding supported by prior studies[24,25]. Pronounced clustering of MSM and MSW gut microbiota community profiles was observed independently of HIV status (Adonis, $P = 0.001$), suggesting that sexual practice per se may drive variability in bacterial composition (Fig. 3b). We next compared abundances of ASVs in MSM vs. MSW (Supplementary Data 5), and found that MSM were characterized by significantly higher abundance of several *Prevotellaceae* members in agreement with previous reports[24–26], as well as increases in *Coriobacteriaceae*, *Erysipelotrichaceae*, and *Clostridiales* (primarily *Lachnospiraceae* and *Ruminococcaceae*) members (Fig. 3c). Conversely, frequencies of *Bacteroides* and *Rikenellaceae* members as well as *Akkermansia muciniphila* were lower in MSM compared to MSW (Fig. 3c).

Prior studies examining the gut microbiota of PWH may have been confounded by an over-representation of MSM in the PWH grouping as compared to a seronegative control grouping drawn from the general population in which MSM may be a minority. Thus, taxa enriched in MSM due to the MSM-associated

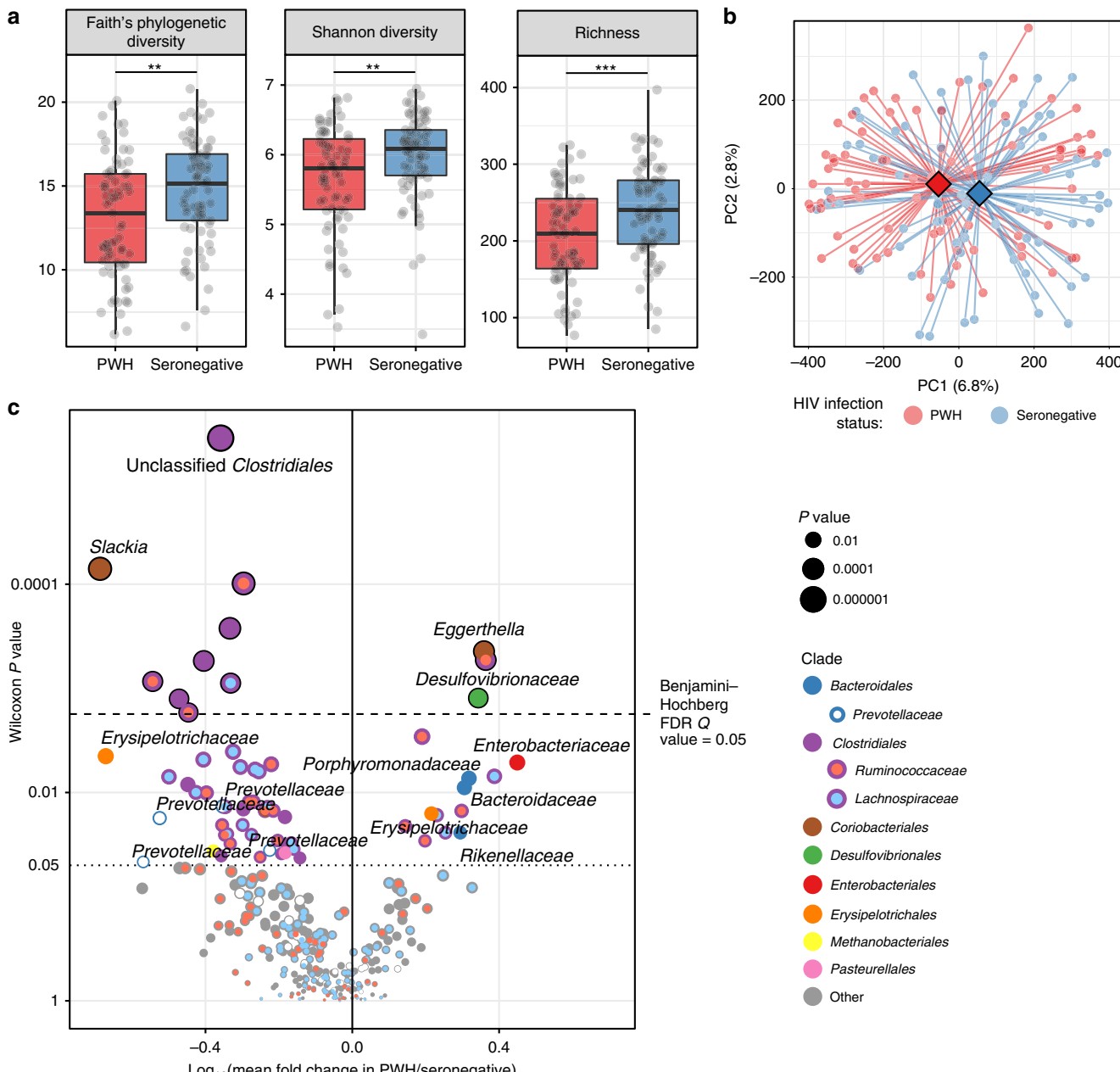

**Fig. 1 Gut microbiota composition differs between PWH and uninfected controls. a** Alpha diversity measures are shown for all participants, grouped as PWH and seronegative controls ($n = 142$, two-tailed paired $t$-test $P = 0.0012$, $P = 0.0012$, $P = 0.0004$, respectively). Boxes denote inter-quartile range, bar denotes median, and whiskers denote range. **b** Principal coordinates analysis (PCoA) plot based on the Canberra beta diversity metric with centroids depicted as diamonds for PWH and seronegative controls. Community differences were verified by PERMANOVA, $P = 0.001$. **c** Volcano plot depicting ASVs (operational taxonomic units, referring to dada2 sequence variants) in differential abundance between PWH and seronegative controls using the paired non-parametric Wilcoxon signed-rank test, with pairs of PWH and controls matched for sex, sexual practice, age, BMI, and birth country. Benjamini–Hochberg false discovery rate $Q$-value cutoff of 0.05 is shown. ASVs are colored by either their family (for *Lachnospiraceae*, *Ruminococcaceae*, and *Prevotellaceae*) or order (all else), and families of selected taxa are denoted by text. *$P < 0.05$, **$P < 0.005$, ***$P < 0.001$.

microbiota signature may have emerged as false-positive observations conflated with taxa enriched in HIV infection. In order to understand whether the MSM-associated microbiota signature overlaps with and thus explains the appearance of an HIV-associated microbiota signature, we sought to address the important unanswered question of whether taxa observed to be enriched in PWH are the same as those enriched in MSM or conversely whether those depleted in PWH were also depleted in MSM. Of ASVs that differed significantly between PWH vs. seronegative (total ASVs = 65 with $P < 0.05$) and MSM vs. MSW

(total ASVs = 99 with $P < 0.05$), 20 ASVs overlapped between these lists (13.9% of the two lists). Surprisingly, these ASVs were found to exhibit opposing abundance trends among these two sets of comparisons (Fig. 3d). Specifically, *Prevotella* taxa and several *Lachnospiraceae* and *Ruminococcaceae* members were enriched in MSM but depleted in PWH. These data reveal that the MSM microbiota signature is complex and unique, and importantly, mostly non-overlapping with that of HIV infection with some taxa exhibiting opposing abundance shifts as those in the HIV microbiota signature.

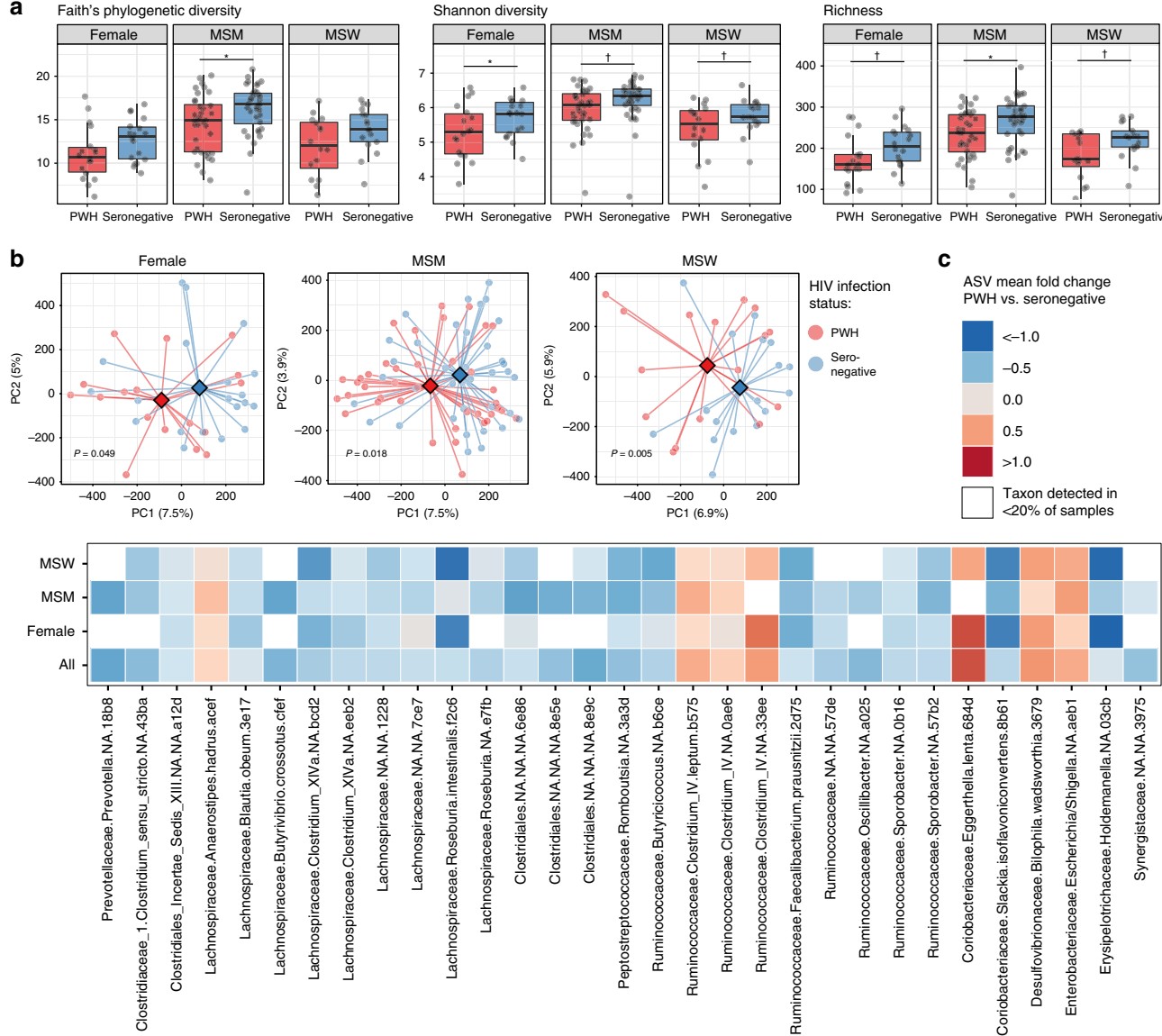

**Fig. 2 Gut microbiota composition differences are evident within subgroups. a** Alpha diversity measures are shown for the three sex/sexual practice subgroups ($n = 36$ females, $n = 72$ MSM; men who have sex with men, $n = 34$ MSW; men who have sex with women), split by PWH and seronegative controls. Paired two-tailed $t$-tests were used to test significance of differences between groups ($P = 0.022$, $P = 0.048$, $P = 0.064$, $P = 0.061$, $P = 0.080$, $P = 0.022$, $P = 0.056$, respectively, for those denoted with bar). Boxes denote inter-quartile range, bar denotes median, and whiskers denote range. **b** PCoA plot based on Canberra beta diversity metric with centroids depicted as diamonds for PWH and seronegative controls. Community differences were verified by PERMANOVA (Female $P = 0.04998$; MSM $P = 0.005$; MSW $P = 0.018$). **c** Tile plot showing mean fold change of ASVs among all PWH vs. seronegative participants, with representations of log mean fold change depicted per subgroup for significant ASVs. ASVs with Wilcoxon signed-rank significance $P < 0.0125$ in the comparison of PWH vs. seronegative among all paired participants together (including all three subgroups) are shown. White squares denote ASV being present in fewer than 20% of samples within the given subgroup. $^{†}P < 0.10$, $^{*}P < 0.05$.

**Sexual practice is linked with microbiota regardless of sex.** To explore potential drivers of the MSM-associated microbiota signature, we investigated microbiota trends associated with sexual practice of PWH and HIV-seronegative controls during the six months prior to study visit. Ninety-one percent of MSM completed a behavioral questionnaire that provided information on recent receptive anal intercourse within the last six months (RAI). We found that out of 72 total MSM profiled, 44 reported having engaged in RAI (MSM/RAI+), and 28 reported no such recent activity (MSM/RAI−), with no significant differences in RAI activity between PWH-MSM and SN-MSM ($P = 0.73$). In contrast to two prior studies[25,26], we observed significant community differences between MSM/RAI+ and

MSM/RAI− (PERMANOVA $P = 0.01$, Supplementary Fig. 3). Numerous ASVs were found to be in differential abundance between MSM/RAI+ and MSM/RAI− (Supplementary Data 6), including a depletion of ASVs belonging to the clades *Clostridiales, Veillonellaceae,* and *Akkermansia* in MSM/RAI+, whereas members belonging to the *Prevotellaceae* family were enriched in MSM/RAI+ (Fig. 4a). Next, we selected the top ASVs that differentiated MSM vs. MSW (ASVs with $P < 0.05$ in Fig. 3c) and compared them to ASVs that differentiated MSM/RAI+ vs. MSM/RAI− (ASVs with $P < 0.05$ in Fig. 4a). Among the taxa that overlapped in these two lists, there was a clear consistency in abundance trends such that taxa that were enriched in MSM vs. MSW were also enriched in MSM/RAI+ vs. MSM/RAI− and this

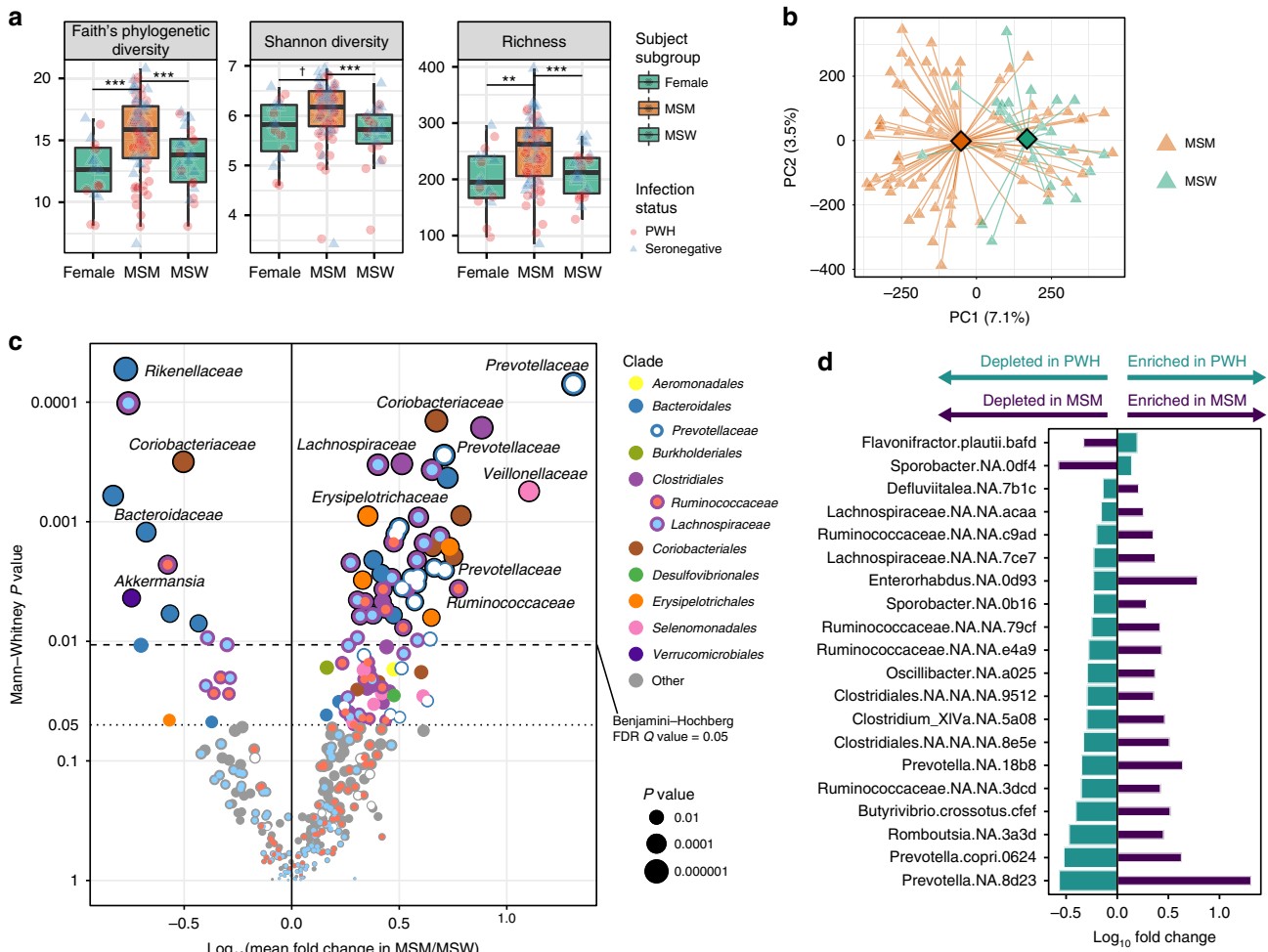

**Fig. 3 Sexual practice exerts major impact on gut microbiota composition. a** Alpha diversity measures are shown for the three sex/sexual practice subgroups ($n = 16$ female, $n = 72$ MSM, $n = 25$ MSW), with participants matched for birth country (Netherlands). Faith's phylogenetic diversity $P = 3 \times 10^{-7}$, Shannon Diversity $P = 2.3 \times 10^{-6}$, Richness $P = 3.2 \times 10^{-7}$ by Kruskal–Wallis tests. †$P < 0.10$, *$P < 0.05$, **$P < 0.005$, ***$P < 0.001$ by Mann–Whitney U-tests. Boxes denote inter-quartile range, bar denotes median, and whiskers denote range. **b** Principal coordinates analysis (PCoA) plot based on Canberra beta diversity metric with centroids depicted as diamonds for MSM and MSW. Shown are participants matched for birth country (Netherlands) and infection status between the comparator groups of MSM and MSW ($n = 72$ MSM [36 PWH and 36 seronegative] vs. $n = 22$ MSW [11 PWH and 11 seronegative]). Community differences were verified by PERMANOVA, $P = 0.001$. **c** Volcano plot depicting ASVs in differential abundance between MSM and MSW matched for infection status and birth country, using the unpaired non-parametric Mann–Whitney U test. ASVs are colored by either their order or family, and families of selected taxa are denoted by text. **d** A comparison of abundance trends among ASVs significantly differing in both the comparison of MSM vs. MSW and PWH vs. uninfected controls. ASVs were selected by P-value < 0.05 in both comparisons (paired Wilcoxon for PWH vs. uninfected, unpaired Mann–Whitney for MSM vs. MSW). Directionality of ASV abundances are in opposition to each other and refute the expectation that having MSM status enriches for the same taxa as does HIV infection (Fisher's exact test $P < 0.001$).

pattern was mirrored in the depleted bacterial taxa ($P = 0.006$, Fisher's exact test, Fig. 4b). The bacterial communities increased in MSM and MSM/RAI+ included *Prevotella*, *Collinsella*, and *Oribacterium* members, whereas *Blautia faecis* and *Akkermansia muciniphila* were depleted in both MSM and MSM/RAI+. To further examine sexual practice-specific changes in the composition of the gut microbiota, we sought to examine the role of receptive anal intercourse in female participants within our cohort. Using the Canberra beta diversity distance metric, which quantifies the overall ecological community similarity between two microbiota samples and is well-suited for microbiota count data[31], we quantified the similarity of gut microbiota profiles of females that reported RAI (F/RAI+, $n = 5$) to MSM that reported RAI (MSM/RAI+), as well as that of females without RAI ($n = 33$) to MSM/RAI+ participants. We found that females with RAI exhibited greater bacterial community similarity to the MSM/RAI+ population than did females that reported no RAI ($P < 0.05$)

(Fig. 4c). Consistent with the impact of RAI in males, all ASVs that shifted in both females with RAI vs. females without RAI and MSM vs. MSW did so in the same direction ($P = 0.033$, Fig. 4d). Finally, we found that when comparing the overall community structure of MSM that engaged in recent condomless RAI ($n = 31$) to that of MSM that engaged in RAI with use of a condom ($n = 18$), there was not a significant difference in community composition ($P = 0.16$, PERMANOVA). In summary, these data reveal that sexual practice is associated with a unique microbiota signature regardless of sex and possibly regardless of condom use.

**Robustness of MSM- and HIV-associated microbiota signals.** Microbiota survey studies comparing diseased vs. healthy individuals are highly prone to false discovery due to the high numbers of comparisons being made (~$10^3$ taxa commonly

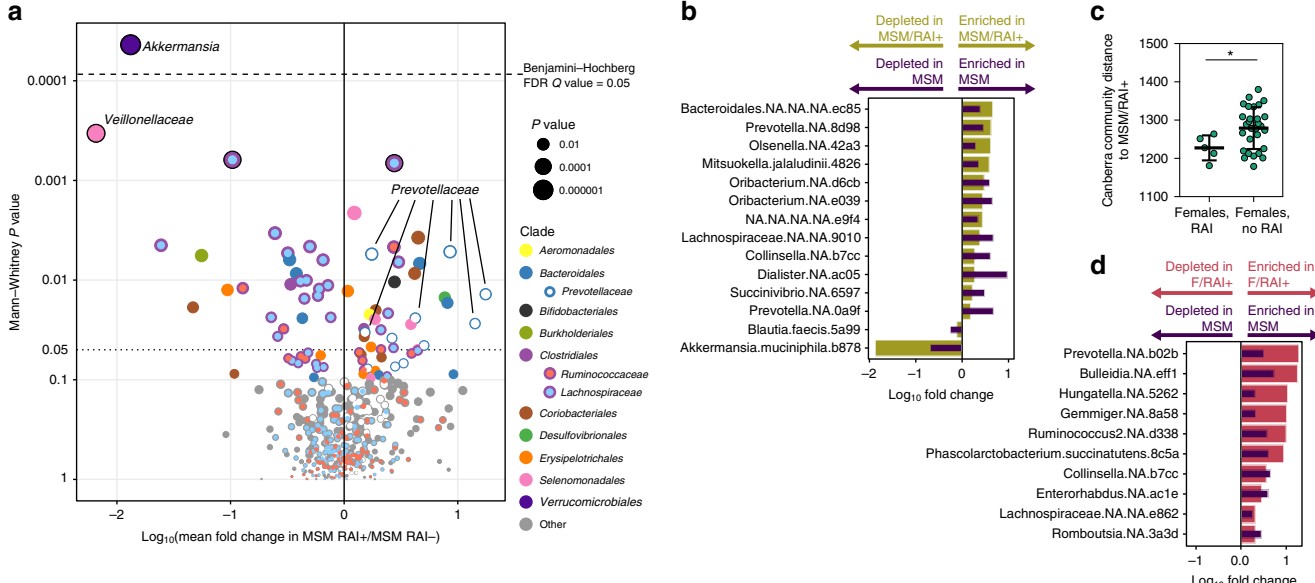

**Fig. 4 Recent sexual practice and lifetime sexual practice impacts gut microbiota composition regardless of sex. a** Volcano plot depicting ASVs in differential abundance between MSM/RAI+ (men who have sex with men that experienced receptive anal intercourse within the last 6 months prior to sampling, n = 44) and MSM/RAI− (men who have sex with men that did not experience receptive anal intercourse within the last 6 months prior to sampling, n = 28), matched for infection status. **b** Overlap in abundance trends among the top 100 taxa in differential fold abundance among MSM vs. MSW (n = 72 MSM, n = 22 MSW) and MSM/RAI+ vs. MSM/RAI− (n = 44 MSM/RAI+, n = 28 MSM/RAI−). Mann–Whitney tests were performed on each ASV for both comparisons, and shown are taxa that reached P < 0.05 on both lists. **c** Average Canberra community distances to MSM/RAI+ were calculated for each female participant sample that reported receptive anal intercourse within 6 months prior to sampling (F/RAI+, n = 5) and female participants that reported no such activity (F/RAI−, n = 28). F/RAI+ gut communities were significantly more similar to MSM/RAI+ than were those for F/RAI−. Unpaired two-tailed t-test P = 0.0496. Middle bar denotes average and error bars represent SD. **d** Overlap in abundance trends among the taxa that reached P < 0.05 in both comparisons of MSM vs. MSW and F/RAI+ vs. F/RAI− (Fisher's exact test P = 0.077), as performed in **b**.

compared across groups) and the subsequently high likelihood of random observations appearing significantly discriminative of cases and controls. One method of overcoming the challenge of identifying signal among the noise is to examine the consistency of taxonomic shifts across independent studies, an exercise that largely supports the shifts reported herein[15–22,24]. Machine learning offers another attractive methodology to reduce false discovery using the concept of intra-study cross-validation. By this method, a binary classification model identifies the most discriminatory bacterial taxa between positive and negative classes (i.e. cases and controls) from a subset of samples—known as the 'training' set. Then, the capacity for the model to accurately predict the class of the remainder of samples—'validation' set of unseen samples held-out from the training of the model—is determined. Models are repeatedly trained on randomly selected training subsets and evaluated on the remaining held-out validation data. The overall capacity for the gut microbiota to accurately predict unseen samples into their correct class is quantified numerically as the area under the curve of the receiver-operating characteristic curve (AUROC), which can be interpreted as the probability that, given a random pair of unseen positive and negative samples, the classifier will choose the positive sample over the negative sample as belonging to the positive class. In this manner, we utilized ridge logistic regression binary classifiers to estimate the overall robustness of the gut microbiota to predict MSM vs. MSW and PWH vs. HIV-uninfected, separately. In both cases, the microbiota was able to classify persons with significant prediction accuracy with AUROC = 0.84 ± 0.07 (P = 0.015) for MSM vs. MSW and AUROC = 0.73 ± 0.06 (P = 0.017) for PWH vs. seronegative (Fig. 5a), indicating that both HIV infection status and MSM status had robust microbiota signatures and that the MSM

microbiota signature was more robust than that of HIV. This is consistent with the observation that a greater number of ASVs differed significantly between MSM vs. MSW as compared to PWH vs. seronegative (Supplementary Data 4 and 5). Results were similar when using random forests binary classifiers (Supplementary Fig. 4). To test whether our model was producing accurate predictions due to true microbiota differences between our comparator rather than overfitting to noise within the taxon abundance data, samples were randomly assigned to either MSM or MSW in the MSM vs. MSW comparison, or separately to PWH or seronegative for its respective comparison, and our pipeline was performed on the resulting corrupted data. When sample class labels were thus randomized, results were consistent with guessing near random chance (0.50 ± 0.13 and 0.50 ± 0.08, for MSM vs. MSW and PWH vs. HIV-seronegative, respectively, Fig. 5b), supporting the results of the true groupings and the overall conclusion that both sexual practice and HIV infection status exert influences on the gut microbiota that are non-random, robust, and consistent across samples.

Ridge logistic regression models allow for the use of feature parameter coefficients (i.e. feature importances) that indicate the relative association of each ASV and the classes, learned by the model during minimization of the classification error (Supplementary Data 7 and 8). ASV features that were most informative for predicting PWH vs. HIV-seronegative included *Enterobacteriaceae* and the *Desulfovibrionaceae* member *Bilophila wadsworthia* as being indicative of PWH, and *Coriobacteriaceae* and *Lachnospiraceae* members as indicative of HIV-seronegative samples (Fig. 5c), observations that were largely consistent with those found to be in differential abundance by traditional statistical methods (Fig. 2c) and prior studies[15–22,24]. Interestingly, ASVs that were depleted in MSM had greater capacity to

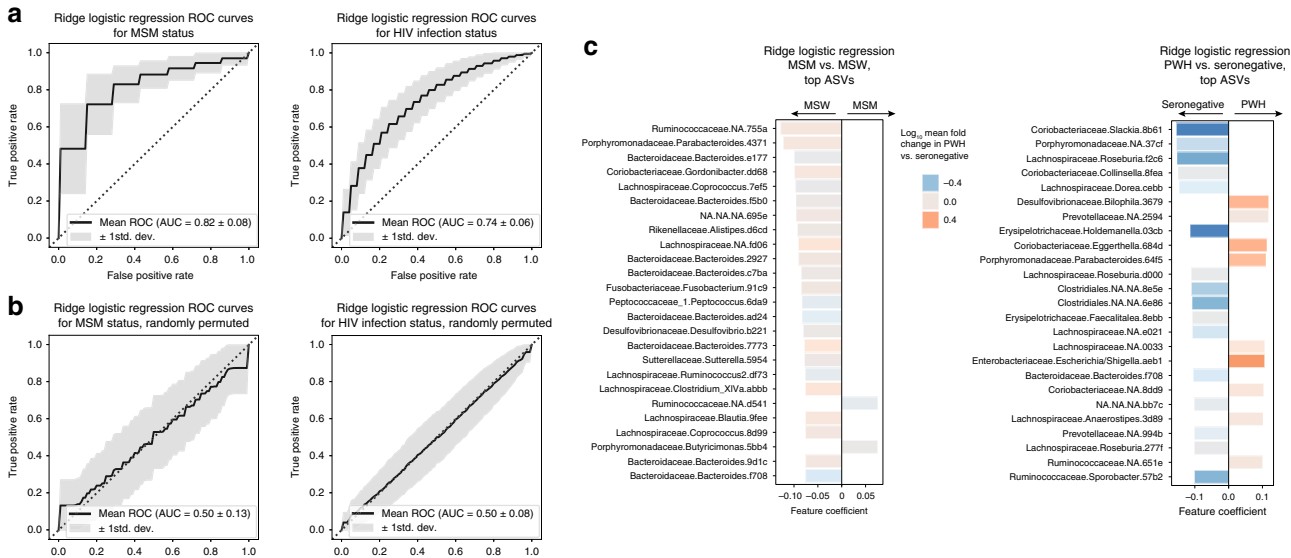

**Fig. 5 Gut microbiota profiles robustly segregate samples by HIV infection status and sexual preference. a** The ridge logistic regression application of machine learning was used to test robustness of differences between MSM vs. MSW ($n = 72$ MSM, $n = 22$ MSW), and PWH vs. uninfected ($n = 71$ PWH, $n = 71$ seronegative). Both MSM status and HIV infection status exhibited robust signatures by which ridge logistic regression could predict the classification status of unseen samples with high accuracy (AUROC = 0.84 for MSM vs. MSW with $P = 0.015$, AUROC = 0.73 for PWH vs. seronegative with $P = 0.017$). **b** Sample classifications (e.g. MSM, MSW, PWH, seronegative) were randomly permuted to introduce noise into the dataset and quantify capacity for machine learning to classify persons based on spurious data. Random class label permutation demonstrated no classification accuracy to spurious data (AUROC = 0.5 for both MSM vs. MSW and PWH vs. uninfected). **c** Top ASV features that informed models for ridge logistic regression predictions of MSM vs. MSW and PWH vs. seronegative. Feature coefficients represent extent to which each ASV was weighted to represent the given class (positive for MSM vs. MSW indicated ASV was indicative of MSM; positive for PWH vs. seronegative was indicative of PWH). Bars are color-coded by log mean fold change in PWH vs. seronegative, and show a lack of correlation between ASVs informative for predicting MSM status and those altered in abundance in PWH.

predict MSM vs. MSW status (Fig. 5c), and these included *Bacteroides*, the *Alistipes* genus of the *Rikenellaceae* family, which have previously been reported as depleted in HIV infection among studies for which a control population was not actively matched for sexual practice[32].

**HIV-associated microbiota correlates with inflammation**. To understand the clinical implications of HIV-associated gut microbiota perturbations, we calculated a 'dysbiosis index' (DI) to collapse into a single number the shifts in bacterial taxa that were characteristic of PWH vs. seronegative participants. This was accomplished by calculating the log ratio of geometric means of taxon abundances that were enriched in PWH (having Wilcoxon $P < 0.05$) over taxa that were depleted in PWH (having Wilcoxon $P < 0.05$) as compared to seronegative controls. The resultant DI differed between HIV-infected groups and was lowest among PWH-MSM (Supplementary Fig. 5), possibly due to abundance trends of certain taxa in the MSM signature being in opposition to the abundance trends for the same taxa in the HIV microbiota signature (Fig. 3d), as well as the generally increased alpha diversity in MSM (both PWH and seronegative, Supplementary Fig. 1a, b) which is contrary to the trend in PWH (Fig. 2a). We next examined associations between DI and clinical characteristics and inflammatory markers within PWH using linear mixed effects models with subgroup (F/MSM/MSW) included as a random effect. We found a strong, significant correlation between the DI and known duration of HIV infection, duration of anti-retroviral therapy (which correlated with duration of HIV infection), as well as nadir CD4 count[33] and pre-ART CD4 count[34] (Fig. 6a, Supplementary Data 9). As reduced alpha diversity was also a feature of the HIV-associated microbiota signature, we performed the same association analysis within

PWH on Shannon diversity vs. clinical and inflammatory markers to find associations with soluble CD14 (sCD14) (Supplementary Fig. 6a, b) and pre-ART CD4 count (Fig. 6b, Supplementary Data 10).

To expand understanding of links between the HIV-associated microbiota and host immune states, we performed unbiased aptamer-based proteomic screens to quantitate 1305 human plasma proteins on a subset of samples evenly split between studied groups (9 SN-F, 9 SN-MSM, 9 SN-MSW, 16 PWH-F, 22 PWH-MSM, and 16 PWH-MSW). Comparison of all plasma proteins between paired PWH and seronegative controls identified several inflammatory markers enriched in HIV infection (Supplementary Data 11) including myeloperoxidase (MPO), growth/differentiation factor 15 (GDF15), and secreted urokinase plasminogen activator surface receptor (suPAR), a marker of systemic immune activation recently identified as a remarkably strong independent predictor of myocardial and overall non-AIDS morbidity and mortality[35,36]. Notably, suPAR was both significantly higher in PWH ($P = 2.6 \times 10^{-5}$, $Q = 0.011$, Fig. 6c, $n = 27$ matched pairs) and inversely correlated with Shannon diversity ($P = 0.0007$, $Q = 0.021$, Fig. 6d) when examining all PWH queried via proteomics ($n = 54$ PWH). Although DI and Shannon diversity were themselves inversely correlated with each other, multivariate linear mixed effects analysis revealed that each was associated with the aforementioned markers of disease independently of the other (CD4 nadir vs. DI and Shannon: DI $P = 0.0173$, Shannon $P = 0.8147$; sCD14 vs. DI and Shannon: DI $P = 0.737$, Shannon $P = 0.0436$; suPAR vs. DI and Shannon: DI $P = 0.981$, Shannon $P = 0.0088$). Furthermore, suPAR positively correlated with the DI ($P = 0.0987$, Supplementary Fig. 6c) though did not reach statistical significance, and fold enrichment/depletion patterns of ASVs that

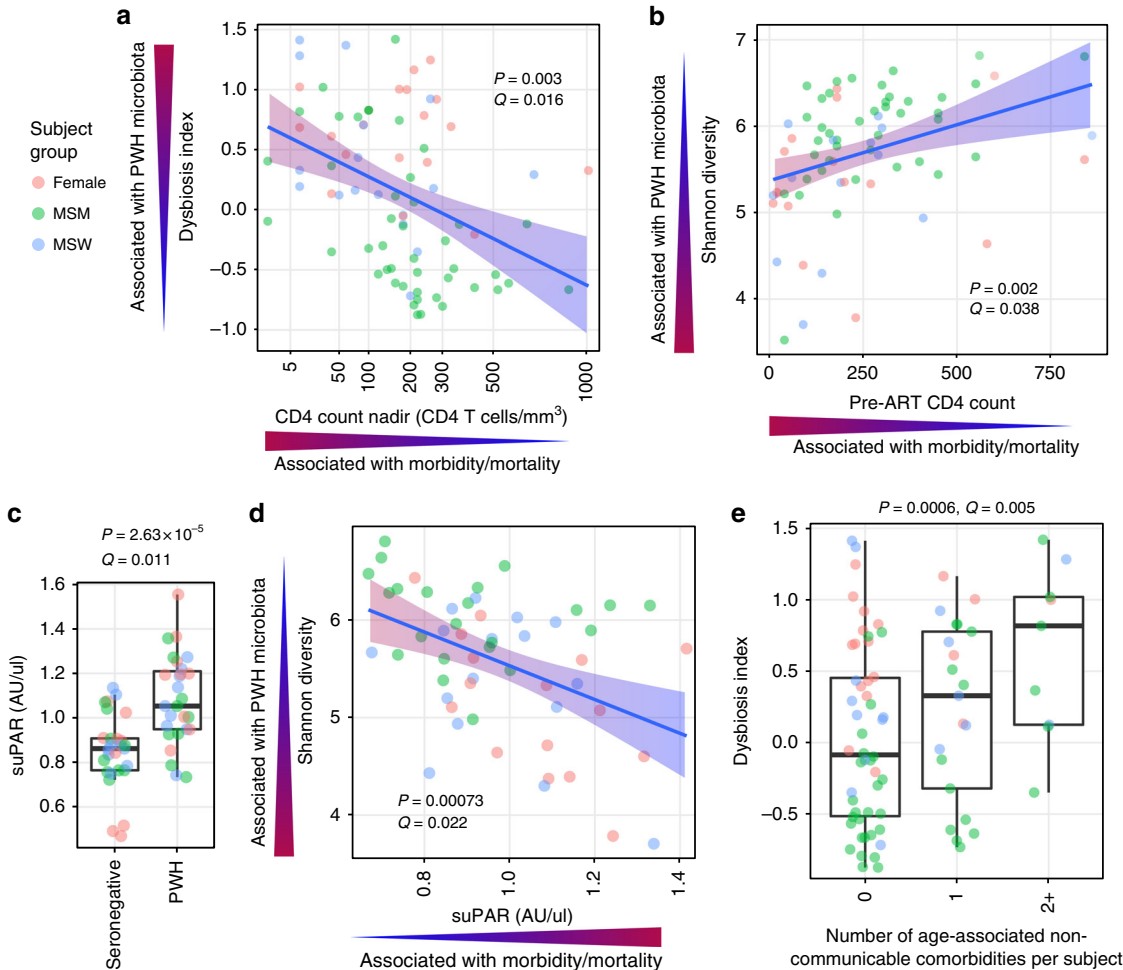

**Fig. 6 HIV-associated gut microbiota features correlate with markers of HIV disease progression, immune activation, as well as comorbidity prevalence.**
Linear mixed effects (LME) models were used to control for subgroup (F, MSM, MSW) as random effects for all analyses depicted. Linear regression lines denoted in blue with 95% confidence intervals depicted as a gradient with red representing directions for both independent and dependent variables that are associated with the PWH state (e.g. high dysbiosis index, low nadir CD4 count) and blue representing the seronegative state. **a** 'Dysbiosis Index' (DI) correlates with nadir CD4 (count/mm³), LME P = 0.003, Benjamini–Hochberg Q = 0.017, among PWH subjects (n = 80). **b** Shannon alpha diversity correlates with pre-ART CD4 count, LME P = 0.002, Q = 0.038. **c** suPAR concentrations in plasma are strongly elevated in PWH (n = 27) compared to matched seronegative controls (n = 27) by paired Wilcoxon signed-rank test. Benjamini–Hochberg Q-values calculated with consideration of all 1305 plasma markers quantitated by aptamer-based SomaScan assay. Boxes denote inter-quartile range, bar denotes median, and whiskers denote range. **d** Shannon alpha diversity correlates with plasma suPAR levels, LME P = 0.00073, Q = 0.022, n = 54 PWH. **e** Number of comorbidities experienced by PWH participants (n = 80) by time of sampling correlates with DI, LME P = 0.0006, Q = 0.005.

differed in abundance between PWH and seronegative controls (those comprising the DI) were associated with directionality of correlations with suPAR, such that bacteria depleted in PWH correlated inversely with suPAR levels in PWH, whereas bacteria enriched in PWH correlated positively with suPAR levels (Supplementary Fig. 6d), suggesting a link not only with alpha diversity but also with the specific taxa that are altered in abundance in PWH.

**Comorbidity prevalence is linked with HIV-associated microbiota.** Having found links between clinical and inflammatory markers of disease progression with microbiota features, we sought to test whether microbiota features associated also with inflammatory comorbidity events including type II diabetes, renal disease, fibrosis (using FIB-4), osteoporotic fractures, chronic obstructive pulmonary disease (GOLD 2 and higher), diagnosed myocardial infarction, cerebrovascular accident, transient ischemic attack, angina pectoris, peripheral arterial disease, and

non-AIDS associated cancer, which all occurred in a subset of our treated PWH cohort at varying prevalence and to a greater degree than in seronegative controls (Supplementary Fig. 7). The DI, though not Shannon diversity, was significantly associated with the number of comorbidities (0,1, ≥2) (Fig. 6e), and this effect was observed within individual subgroups (Supplementary Fig. 8). Furthermore, DI retained a significant association with comorbidity prevalence when including CD4 nadir as a co-variate in a linear mixed effects model framework (DI P = 0.0044, CD4 nadir P = 0.4523).

## Discussion

The role of enteric microbiota alterations in chronic HIV infection has been the focus of increasing investigation. Chronic, systemic immune activation is a hallmark of treated and untreated HIV infection that is considered the primary driver of viral replication and disease progression[37,38]. Presence of specific gut microbiota members in mice has been causally implicated in

spurring pathologic inflammation in experimental models[7,39], leading to the hypothesis that dysregulation of the gut microbiota in humans may also contribute to human inflammatory diseases such as chronic HIV infection. Although many studies have found alterations in the gut microbiota of HIV-infected individuals including enrichment of bacteria known to be pro-inflammatory and a simultaneous depletion of tolerogenic, homeostasis-promoting bacteria, recent seminal studies have revealed a major shift in the gut microbiota of MSM as compared to MSW. Such reports have called into question earlier observations of microbiota dysregulation in relation to HIV per se[24,40]. Of note, numerous early studies recruited primarily MSM in the PWH cohort and did not actively match cases and controls for sexual practice, causing an enrichment of MSM in PWH as compared to uninfected control comparators. Due to the existence of such confounding variables and to the tremendous heterogeneity in gut microbiota profiles within the human population, the lack of a well-powered cohort that simultaneously examines the impact of HIV infection in females and males with differing sexual practices while being carefully matched for microbiota-modulating confounding variables has hampered our understanding of HIV-associated dysbiosis and its role in HIV disease. Furthermore, a lack of precise delineation of how the MSM-associated microbiota compares or contrasts with that of HIV infection remains a barrier toward identifying gut microbiota constituents linked to HIV disease.

In this study, we dissect the differential impacts on the gut microbiota for two major sources of microbiota heterogeneity: that of MSM status and HIV infection status. We find gut microbiota shifts associated with HIV infection in our comparison of antiretroviral-treated PWH as compared to uninfected controls carefully matched for age, sex, sexual practice, birth country, and BMI. Alpha diversity shifts as well as differences in relative abundance of several taxonomic clades of bacteria were found to have consistency across three major groups of interest: females, MSW, and MSM. Components of HIV-associated microbiota shifts correlated with clinical and inflammatory markers of HIV disease progression as well as prevalence of age-associated noncommunicable comorbidities, highlighting the need to elucidate causality in these associations. Furthermore, we characterize a robust microbiota signature associated with MSM status that is consistent with prior reports[24–26]. Importantly, we build upon these observations to contrast the microbiota signatures of MSM status and HIV infection status to show that the MSM-associated microbiota signature is unique from that of HIV infection, and that taxa that constitute both the MSM microbiota signature and the HIV microbiota signature exhibit opposing abundance trends. Finally, we provide evidence that the MSM-associated microbiota signature may be driven by receptive anal intercourse (RAI), a hypothesis supported by our examination of RAI in females.

Our results reveal that antiretroviral-treated HIV infection is associated with distinct gut microbiota features independently of sex/sexual practice, in the form of a reduction in alpha diversity, overall shifts in community structure as denoted by beta diversity analyses, and consistent shifts in abundance of specific bacterial taxa common across participant groups. The bacterial shifts observed in PWH included an enrichment of pro-inflammatory taxa belonging to *Enterobacteriaceae* (*Escherichia/Shigella*) and *Desulfovibrionaceae* (*Bilophila wadsworthia*) consistent with other cohorts[16,22,24]. Notably, the *Enterobacteriaceae* family includes several pathogenic taxa such as *Escherichia, Shigella,* and *Klebsiella* which are well documented sources of bacteremia in PWH. Members of the *Desulfovibrionaceae* family have also been implicated in chronic inflammatory states such as inflammatory bowel disease[28]. *Desulfovibrionaceae* encompass several species

with the metabolic capability to reduce sulfate which produces the byproduct hydrogen sulfide, a cytotoxic compound that inhibits colonocyte butyrate oxidation. Butyrate is a crucial energy source for colonocytes, an inducer of inter-enterocyte tight junction proteins that promote gut barrier function[41], and augments gut regulatory T cells which constitute a lymphocyte population critical for mitigating pathologic inflammation[42,43]. A concomitant decline in butyrate-producing gut microbiota members belonging to the *Lachnospiraceae* and *Ruminococcaceae* families[44] was observed in PWH compared to HIV-seronegative controls. Indeed, abundance of *Lachnospiraceae* members in HIV-infected people has been recently found to correlate inversely with I-FABP, a marker of gut enterocyte turnover, supporting the importance of this clade in mucosal homeostasis[45], though I-FABP did not correlate with our dysbiosis index in this study. Although the enterocyte turnover that is measured indirectly by serum I-FABP may represent loss of mucosal integrity in cases of enterocyte death[46], reduced expression of tight junction proteins between enterocyte cells can also increase likelihood of microbial translocation. Tight junction expression is positively regulated by butyrate[41] and thus may constitute a pathway by which the HIV-associated microbiota may impair gut barrier function independently of enterocyte death, via lessened butyrate production. Thus, an increase in production of molecular inhibitors of butyrate oxidation in combination with a decrease in butyrate-producing gut bacteria may synergize to contribute to the epithelial barrier disruption that is seen to persist in treated HIV infection[10]. This impairment of the function of the intestinal epithelium may permit translocation of pro-inflammatory microbiota products into systemic circulation and exacerbate pathological chronic inflammation in treated PWH[47–50], providing a putative link between the described HIV-associated microbiota features and noncommunicable, inflammation-associated comorbidities.

The extent of dysbiosis in our study was found to correlate with CD4 nadir and pre-ART CD4 counts, both being measures of the severity of immune collapse during untreated infection which suggests a link between immune function and preservation of microbial community composition. Dysbiosis was tightly correlated with the prevalence of comorbidities, which themselves have previously been linked to CD4 nadir and pre-ART CD4 counts[33,34]. It is possible that the extent of immune damage (proportional to the extent of CD4 T cell loss, i.e. CD4 nadir) dictates the degree of dysregulation of microbiota composition but is itself the cause of later age-associated comorbidities, with the effects on the microbiota contributing little to pathology. However, extent of dysbiosis was significantly correlated with comorbidities independently of CD4 nadir, suggesting that the link between dysbiosis and disease progression is not merely an innocuous consequence of prior events but may impact later comorbidity occurrence. Indeed, innate immune activation may constitute a link between the comorbidities measured herein— each of which have been linked to innate immune activation levels in PWH[51–55]—and the gut microbiota, which has been shown to causally impact innate immune activation states in mice[7,39]. SuPAR is a marker of innate immune activation[56], which has been recently characterized as a robust predictor of myocardial[35] and overall non-AIDS clinical events (with an odds ratio of 2 per inter-quartile range for death[36]) in PWH. It has also been described to be an accurate predictor of bacteremia[57,58] and is increased as a result of LPS injection in humans[59] but does not increase in certain cases of sterile inflammation[60], prompting the hypothesis that its capacity to predict HIV-associated mortality may be due to its relationship with bacterial translocation. Our finding that suPAR levels are linked with features of the HIV-associated microbiota signature (i.e. overall alpha diversity and

specific taxa associated with PWH) may prompt its usage as a biomarker for microbiota-mediated strategies to alleviate HIV-associated inflammation and warrants further study into the impact of HIV-associated gut bacteria on this pathologic inflammatory pathway.

We report that MSM harbor a markedly different gut microbiota composition than that of MSW. Though HIV infection status exhibited a strong microbiota signature, the effect of MSM on the microbiota was more pronounced than that of HIV status by both numbers of ASVs that were in differential abundance and by robustness as assessed via machine learning analyses. We find overlap between bacterial community shifts in MSM and MSM/RAI+, supporting the idea that RAI represents a driving factor for microbiota differences observed between MSM and MSW. The lack of bacterial community differences between condomless MSM/RAI+ and MSM/RAI+ with condom suggests that the RAI-associated microbiota may not be an effect of semen but perhaps is linked with other practices associated with RAI, for example rectal douching and use of lubricants. Interestingly, the females who reported engaging in RAI bore compositional similarities to MSM/RAI+, indicating that RAI may be linked to a microbiota signature regardless of gender. Although we found statistically significant results in this analysis, our sample size was limited and further study is warranted. The RAI-associated microbiota in both MSM and F was mainly characterized by an expansion in the *Prevotella* genus which has been previously associated with lubricant use[61]. A decline in abundance was observed for *Bacteroides* members as well as *Akkermansia muciniphila*. Both *Bacteroides* members and *A. muciniphila* are known to utilize mucus as major growth and adhesion substrates that can reside preferentially in the mucus layer of the large intestine[62,63]. These observations prompt the hypothesis that the colonic mucus layer of MSM undergoes remodeling and possible depletion. Further investigation is warranted as to the consequences of RAI on host mucus integrity and on the microbiota, as well as the effect of these microbial community shifts on host physiology, especially given recent studies showing that *Akkermansia* supplementation in humans promotes mucosal barrier health with the effect of reducing plasma LPS levels and ameliorating metabolic syndrome[64]. Furthermore, due to the robust nature of the MSM-associated microbiota signature, we recommend that sexual preference data be systematically collected and taken into consideration in microbiota studies beyond HIV.

Earlier cross-sectional studies examining the gut microbiota in PWH that were not matched for microbiota-associated confounding variables such as MSM status (primarily having an over-representation of MSM in the PWH group) reported shifts from *Bacteroides* to *Prevotella* predominance following HIV-1 infection[18–23]. Stratifying participants by sex/sexual practice, we found that enrichment of *Prevotella* and depletion of *Bacteroides* were not features of HIV infection per se but were instead characteristics of the MSM-associated microbiota. However, we found evidence that treated HIV infection does exert a distinct effect on the microbiota that is largely distinct from that of MSM status and is characterized by a reduction in overall diversity, enrichment of *Gammaproteobacteria*, and depletion of *Clostridiales* members, a finding strengthened by our having matched cases and controls for metadata variables reported to affect the microbiota (age, BMI[65], birth country[66], and sex/sexual practice[24–26]). Interestingly, bacterial taxa associated with MSM either did not overlap with taxa of the HIV-associated microbiota signature or when overlap did occur (as was the case for 13.9% of the total number of taxa that comprised the MSM and HIV microbiota signatures in our study) these taxa exhibited abundance trends opposite that of HIV infection, supporting the conclusion that some earlier studies examining the gut microbiota

in HIV in which MSM status was not actively matched between cases and controls had a reduced capacity to discern true differences in the microbiota driven by HIV infection and an increased risk of spurious observations linked to MSM status rather than HIV serostatus.

As with all cross-sectional human studies, one cannot rule out the effects of other uncaptured confounding factors that differ across a population. Although diet is known to affect the gut microbiota, HIV gut microbiota survey studies that have examined diet have found evidence that dietary intake in PWH does not correlate with abundances of gut microbes of the HIV-associated microbiota signature[20,24,45] as it does in the seronegative population[20,67]. For example, relative abundance of *Bacteroides* members correlates with meat intake in seronegative subjects[20,67], but this association was not found in PWH[20], suggesting HIV infection may have a greater impact on the abundance of certain microbes than do certain major components of diet. Furthermore, dietary patterns between PWH and seronegative participants did not differ significantly[20,45], suggesting that diet alone is unlikely to be the driver of the HIV-associated microbiota signature. Furthermore, given the potent impact of sexual preference on the gut microbiota, it is conceivable that sexual activities not captured in the present dataset may impact the microbiota signatures described herein, warranting further study. Longitudinal studies may also help disambiguate microbiota signals from confounding factors, though some lifestyle factors may change with diagnosis and should likely be continuously monitored. Relative longitudinal stability of the gut microbiota also was not examined in the present study, though prior studies have noted limited intra-individual dispersion among PWH[68,69]. We cannot exclude that HIV infection induces microbiota instability in humans, which was not able to be captured in the present cross-sectional study design. An added and as-yet unaddressed caveat to the present study is the inability to tease apart the role of HIV infection from that of antiretroviral therapy, as all our PWH participants were also treated over a long term (median of 12.5 years). Though several studies report gut microbiota shifts in untreated persons that bear similarities to those reported herein[15–18,20,23], the effect of antiretrovirals on the gut microbiota remains poorly understood and cannot be ruled out as a contributor to the observed microbiota shifts in the absence of studies directed at this question. Furthermore, although country of birth and immigration status were collected in our metadata, ethnicity information was not and may be an important potential confounding factor to collect in future studies.

Avenues to further elucidate the role of the microbiota in HIV disease progression include the examination of mucosal-resident communities in the context of the populations studied herein (i.e. MSM, PWH). Given the mucosal-resident nature of potently immunostimulatory SFB and *Akkermansia* in mice, and given that this community differs from that of the lumen, studies examining microbial communities of mucosal biopsies may yield important insights into the relationship between the gut microbiota and immune states in HIV infection. Non-human primate studies offer an attractive path for longitudinal analysis, though challenges in translating SIV microbiota studies exist in that SIV infection does not durably alter the macaque gut microbiota through the early phases of chronic infection[70–72]. However, wild chimpanzee studies have reported microbiota instability in advanced SIV infection[73,74], and experimentally infected rhesus macaques similarly exhibit microbiota alterations during advanced disease characterized by low CD4 counts[75] not unlike those described herein (i.e. *Enterobacteriaceae* bloom). Given that nadir CD4 count correlates in our study with extent of dysbiosis, it is possible that the inflammation and/or immune destruction

occurring in advanced HIV and SIV disease causes microbiota disruptions that persist even after CD4 restoration via treatment, and that this altered microbiota contributes to elevated comorbidity burden in PWH. Future prospective longitudinal studies in humans and non-human primates are warranted to further dissect the impact of the observed specific shifts in the gut microbiota on associations with chronic inflammation and comorbidity prevalence in HIV as reported herein.

## Methods

**Study participants and study design.** The AGE$_h$IV Cohort Study is an ongoing observational cohort (NCT01466582) of HIV-infected participants from the HIV outpatient clinic of the Amsterdam University Medical Centers (location Academic Medical Center) and HIV-uninfected participants recruited from either the sexual health clinic or the Amsterdam Cohort Studies on HIV/AIDS at the Public Health Service in Amsterdam, the Netherlands. All participants signed informed consent prior to study procedures. Participants were 45 years or older at enrollment and attended biennial study visits. Participants were requested to complete a standardized questionnaire concerning demographics, personal and family medical history, use of medications, participation in population screening programs, substance use, quality of life, depression, sexual orientation/behavior/dysfunction, cognitive complaints, calcium/vitamin D intake, physical exercise, social behavior, and work participation/income. During each study visit, participants underwent a venous blood draw, measurements of blood pressure, height, weight, and hip/waist circumference, as well as electrocardiography, spirometry, and bone densitometry. Fecal and urine samples were also cryopreserved during each study visit. Study participants self-collected stool at their homes no more than 24 h prior to study visits and stored specimens in a refrigerator until being brought in for their visits. None of the respondents included in the final analysis reported antibiotics usage at the time of the study visit at which time fecal samples were collected. For this cross-sectional study, one fecal sample was analyzed per individual.

Cryopreserved fecal samples ($n = 169$) were processed from PWH and HIV-seronegative controls that were matched for age, BMI, smoking, and immigration status within each of the three subgroups: men who have sex with men (MSM), non-MSM males (MSW), and females (F). Efficacy of matching for these variables (defined as no significant difference between PWH and seronegative participants within each subgroup) is shown in Supplementary Data 1. The following numbers of samples were successfully extracted, amplified, and sequenced to sufficient depth: 83 MSM, 38 MSW, and 39 females. Paired samples for which the matched sample failed extraction, amplification, or to reach sufficient sequencing depth were removed for paired statistical comparisons. Thus, for paired statistical analyses, numbers of samples were as follows: MSM ($n = 72$), MSW ($n = 34$), and F ($n = 36$).

**Assessment of comorbidity.** Comorbidities objectively assessed during each study visit were: (1) chronic obstructive pulmonary disease (COPD) (GOLD 2 or more by spirometry); (2) advanced liver fibrosis (FIB-4 ≥ 3.25); (3) decreased kidney function (eGFR < 60 mL/min/1.73 m$^2$ using the Chronic Kidney Disease Epidemiology Collaboration [CKD-EPI] equation); (4) osteoporotic fractures (Dual-energy X-ray absorptiometry [DXA] $T$-score < −2.5 standard deviation (SD) for men aged ≥50 years and post-menopausal women or a $Z$-score < −2.0 for men aged <50 years and pre-menopausal women using World Health Organization definitions and having reported a fracture 2 years before or after DXA measurement with osteoporosis as an outcome); (5) diabetes mellitus (HbA1c ≥ 48 mmol/mol or elevated blood glucose (non-fasting ≥11.1 mmol/L or fasting ≥7.0 mmol/L) or using antidiabetic medication). Self-reported comorbidities included: (6) heart failure (diagnosed by cardiologist); (7) non-AIDS defining cancers (confirmed by pathologist, excluding non-melanoma skin cancers); and (8) atherosclerotic disease (diagnosed by specialist; myocardial infarction, angina pectoris, peripheral arterial disease, ischemic stroke, or transient ischemic attack).

Self-reported diagnoses were validated using hospital records for HIV-positive participants, and general practitioners' (GP) records for HIV-seronegative participants who had provided consent to contact their GP. Unvalidated diagnoses were only used when validation was not possible (i.e. participants not providing consent to validate self-reported diagnoses with their GP, or documentation about a self-reported diagnosis was absent for HIV-positive participants who received care in other hospitals during follow-up). Of the 35 self-reported comorbidities, 17 (48.6%) were validated as correct, 16 (45.7%) were rejected, and 2 (5.7%) could not be validated. Prevalence of all aforementioned comorbidities were summed when comparing to Dysbiosis Index and Shannon diversity

**Soluble biomarker quantification.** Soluble CD14 (sCD14), soluble CD163 (sCD163), and I-FABP concentrations were determined in plasma samples stored at −80 °C using enzyme-linked immunosorbent assay (ELISA) (DuoSet ELISAs; R&D Systems) in all 160 subjects. The SomaScan Assay platform was used to measure plasma concentrations of 1,305 human proteins in a subset of $n = 81$ subjects ($n = 27$ seronegative controls and n = 54 PWH; 9 SN-F, 9 SN-MSM, 9 SN-MSW, 16 PWH-F, 22 PWH-MSM, and 16 PWH-MSW). For identification of

plasma proteins in differential abundance between PWH and seronegative controls, we limited analyses only to paired subjects ($n = 27$ seronegative, $n = 27$ PWH, 9 subjects from each subject group per infection status). To mitigate subject group effects, we performed both paired Wilcoxon tests and ran linear mixed effects models using subject group as a random effect. All SomaScan Assay analyte units were normalized and scaled by subtracting the mean value for each analyte and dividing by its standard deviation using the R function 'scale'.

**DNA extraction and PCR amplification.** Bacterial profiles of study participants were generated by broad-range amplification and sequence analysis of bacterial 16S rRNA genes. DNA was extracted from fecal samples using the Qiagen MagAttract PowerMicrobiome DNA/RNA EP Kit automated system in conjunction with the Eppendorf Epmotion 5075 machine. The extracted DNA was then purified using Qiagen DNeasy Blood & Tissue Kit. The V4 hypervariable region of the bacterial 16S rRNA gene was amplified using fusion primers with partial Illumina adaptors. The universal bacterial 515F forward primer (5′-GTGCCAGCMGCCGCGGTAA) and the 806R reverse primer (5′-GGACTACHVGGGTWTCTAAT) were used. PCR reactions were prepared in 100 µL volumes containing 20 µL of 5× Phusion® High-Fidelity (HF) buffer, 2 µL of 10 mM dNTPs, 1 µL of each primer (50 µM), 0.5 µL of Phusion® High-Fidelity DNA Polymerase, and 10 ng of template DNA. Master mixes were then split into triplicate reactions of 33 µL and amplified separately. V4 regions of the bacterial 16 S rRNA gene were amplified using PCR (98 °C for 30 s, followed by 25 cycles at 98 °C for 10 s, 57 °C for 30 s and 72 °C for 30 s, and a final extension at 72 °C for 5 min).

**Illumina MiSeq sequencing.** Following amplification, triplicate samples were pooled and purified using Ampure XP Beads (1:1 ratio). Samples were then quantified using the KAPA qPCR-based Library Quantification kit and pooled at equimolar concentrations. Amplicons were paired-end sequenced (2 × 300 bp) on an Illumina MiSeq platform with a 600-cycle kit using standard protocols.

**16s rRNA sequencing read data processing.** Raw fastq files were processed using the dada2 algorithm (v1.3.3) after primer trimming and trimming amplicon lengths of the forward and reverse reads to 240 bp and 190 bp, respectively. Standard parameters were used (truncQ = 2, maxEE = 2). Taxonomy was assigned to amplicon sequence variants (ASVs) using the RDP training set 16. Fasta sequences for each ASV were processed through the md5sum hash algorithm (R library 'digest' v0.6.23) to create short, unique ASV identifiers. The first four characters of each md5sum ASV identifier are displayed in graphs after taxonomic classifications so as to facilitate linking of 16 s sequences to visualized results. ASVs that were present in fewer than six samples were removed and ASV tables were rarefied to 20,000 reads for subsequent analyses. Phylogenetic trees were constructed and alpha diversity metrics were calculated using QIIME2 (v2018.8).

**Statistical analyses.** For comparisons of overall community similarity, $n$-by-$n$ matrices of the Canberra beta diversity metric (R package 'vegan' v2.5.6) were used in conjunction with the PERMANOVA statistical test (R function 'adonis'). Stratifying by subject group within the PERMANOVA test was performed using the 'strata' argument within the 'adonis' function. To provide adjustments for multiple comparisons in ASV-level comparisons between groups, comparisons of DI and Shannon Diversity to clinical/inflammatory parameters, and proteomic data between groups, Benjamini–Hochberg $q$-values were calculated using the 'p.adjust' function in R. Log mean fold changes were calculated by adding a pseudocount of 1 to all zero-abundance ASVs, calculating mean ASV abundance for each group, and then subtracting the log-transformed means for the two groups from each other. For statistical comparisons that sought to identify ASVs in differential abundance between groups, ASVs were filtered out if present in fewer than 20% of samples to reduce numbers of comparisons for purposes of minimizing false discovery and to filter out rare ASVs for which statistical power to discern differences was low. To identify ASVs in differential abundance between PWH and seronegative subjects, paired Wilcoxon tests were performed for each ASV in series (R package 'exactRankTests' v0.8.30). Paired samples were selected for this analysis, such that each PWH subject was compared to a sample matched for age, BMI, birth country, sex, and sexual preference ($n = 71$ PWH and $n = 71$ seronegative). A two-tailed sampling distribution was assumed for all statistical tests for which a one-tailed or two-tailed distribution could be used. For comparing MSM vs. MSW, participants were selected such that both groups had equal proportions of PWH and seronegative, and were also matched for birth country which was predominantly the Netherlands for this comparison ($n = 22$ MSW and $n = 72$ MSM). Unpaired Mann–Whitney $U$-tests were performed on each ASV to test differences between these two groups. To determine overlap in ASVs that were characteristic of MSM vs. those that were characteristic of PWH, the following was performed. All ASVs that passed a $P < 0.05$ cutoff in the two aforementioned sets of statistical comparisons (PWH vs. uninfected, and MSM vs. MSW) were selected. Those ASVs that overlapped between the resulting lists were then further sub-selected. Fisher's exact test was used to determine whether concordance of abundance trends (same sign of log mean fold changes) among the resulting list of overlapping taxa had occurred by random chance, using an expectation that 50% of taxa should have identical directionality of abundance difference (i.e. same sign of log mean fold changes)

between two random lists as the null hypothesis. The same framework was used to determine consistency of microbiota differences between MSM vs. MSW and MSM/RAI+ vs. MSM/RAI− ($n = 45$ MSM/RAI+ and $n = 33$ MSM/RAI−). First, Mann–Whitney U-tests were performed on each ASV for MSM vs. MSW, MSM/RAI+ vs. MSM/RAI−. ASVs that reached $P < 0.05$ from each of these lists were selected, and those ASVs that overlapped between the resulting lists were then further sub-selected, and Fisher's exact test was used as described above. The same was performed to test significance of the overlap among MSM vs. MSW and F/RAI+ vs. F/RAI− (F/RAI+, $n = 5$ and F/RAI− $n = 33$). To test significance of overlapping trends for taxa that differentiated PWH vs. seronegative in each sex/sexual practice subgrouping (female, MSM, MSW), the top 15 taxa for each sub-grouping list of paired Wilcoxon tests were selected. Fisher's exact test was used as above with the null hypothesis representing the number of taxa that would by random chance exhibit the same directionality of abundance shift in all three groups (25% of taxa). For testing associations involving inflammatory markers, clinical characteristics, prevalence of age-associated noncommunicable comorbidities, and the dysbiosis index (detailed below) or alpha diversity, linear mixed effects analyses were performed using the R packages 'lme4' v1.1.21 and 'lmerTest' v3.1.0. For all such analyses, subgroup (female, MSM, MSW) effects were incorporated into each model as a random effect on intercept (1|Subject Group). When examining associations between two independent variables simultaneously (e.g. dysbiosis index and Shannon diversity) on outcome variables (e.g. clinical, inflammatory marker, or comorbidity prevalence data), both independent variables were treated as fixed effects.

**Dysbiosis index calculation**. Dysbiosis index ('DI') was calculated as the log ratio of geometric means of taxon abundances that were enriched in PWH (having paired Wilcoxon $P < 0.05$ and log mean fold change $>0$ in the comparison of PWH vs. HIV-seronegative controls) over taxa that were depleted in PWH (having paired Wilcoxon $P < 0.05$ and log mean fold change $<0$ in the comparison of PWH vs. HIV-seronegative controls), as follows:

$$DI = \log_{10}\left(\frac{\sqrt[n]{x_1 x_2 \ldots x_n}}{\sqrt[m]{y_1 y_2 \ldots y_m}}\right),$$

where each $x$ denotes read counts for taxa enriched in PWH and $n$ is the total number of such taxa, whereas each $y$ denotes read counts for taxa depleted in PWH and m is the total number of these taxa. Taxon read counts (from a total rarefaction of 20,000 reads per sample) were used with an added pseudocount of 1, in order to accommodate the geometric mean by removing zeroes.

**Data visualization**. Data were visualized using R packages 'ggplot2' v3.2.1, 'ggthemes' v4.2.0, 'RColorBrewer' v1.1.2, 'viridis' v0.5.1, and were formatted for presentation using R packages 'ape' v5.3, 'splitstackshape' v1.4.8, 'reshape2' v1.4.3, 'plyr' v1.8.4, 'dplyr' v0.8.3, and 'grid' v3.5.3.

**Machine learning analyses**. In order to reduce variables for machine learning analyses, ASVs with a mean relative abundance below 0.1% were removed from the ASV table. The ASV table was standard-scaled and log-transformed with a pseudocount of 1. Ridge logistic regression, which reduces the chance of overfitting the training data by penalizing large model weights, was used to classify samples into one of the two binary groups (MSM vs. MSW, or PWH vs. seronegative), with the processed ASV table as the input, predictor data and either MSM and HIV serostatus as the binary response variable. The same subject samples were used for machine learning analyses and traditional statistical analyses described above. PWH participants were matched to seronegative controls for age, BMI, birth country, sex, and sexual preference ($n = 71$ PWH and $n = 71$ seronegative). For MSM vs. MSW, persons were selected such that both groups had equal proportions of PWH and seronegative, and were matched for birth country which was predominantly the Netherlands ($n = 22$ MSW and $n = 72$ MSM). Stratified randomized permutation cross-validation (RPCV) was employed to obtain an accurate estimate of the classification of samples, as well as the ability for the learned ASV associations to be able to classify unseen data. RPCV was used with 500 iterations and 70% of samples used for training and the remaining 30% were used to test the trained model. The area under the curve (AUC) of the receiver-operating characteristic (ROC) curve was used as the measure of model performance on the test set. To demonstrate that the results of the classifier was not just due to random noise within the ASV table, for each iteration of the CV, the response variable was randomly permuted to corrupt the data, and a separate model was trained on the corrupted data in parallel with the model trained on the unpermuted data. The mean-AUC of the corrupted model across all folds was taken as the null hypothesis of the power of the microbiota ASV data to discriminate between the two binary classes and was used to determine the significance of the AUC results of the classifier on the unpermuted data. An alpha of 0.05 was used to determine significance of the performance of the classifiers over the 500 CV iterations.

Performing multiple tests on the same source dataset violates assumptions of the standard t-test (i.e. test and null distributions are inferred from the models trained on the same predictor data), thus an empirical P-value defined by Ojala et al. was used to test if a classifier has found a significant class structure in the data.

The empirical P-value was defined as:

$$p = \frac{|<D \in \hat{D} : e(f, D) \geq \bar{e}(f, D')>| + 1}{k + 1},$$

where $D'$ is the permuted dataset, and $e(f, D)$ is the error of the function $f$ learned on dataset $D$. The empirical P-value thus is taken to represent the fraction of classifiers that performed worse than the averaged permuted classifier performance. Random Forests, a bootstrapped ensemble method of random forest classifiers, was also used to predict samples but underperformed as compared to ridge logistic regression.

**Reporting summary**. Further information on research design is available in the Nature Research Reporting Summary linked to this article.

## Data availability

16s rRNA sequencing data used to generate all figures are available at the NCBI Short Read Archive under accession numbers PRJNA589036 (BioProject) and SRP229524 (SRA). Normalized data for proteomic screens (Supplementary Data 11, Fig. 6, Supplementary Fig. 6) can be found in Source Data.

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

## Acknowledgements

I.V.-C. was funded by the Cancer Research Institute Irvington Postdoctoral Fellowship Award and the National Institutes of Health Intramural AIDS Research Fellowship. This work was supported in part by the intramural research program of NIAID/NIH and by The Netherlands Organization for Health Research and Development (ZonMW) (Grant Number 300020007) and AIDS Fonds (Grant Number 2009063). We thank the AGEhIV Cohort Study group for establishing the cohort from which participants were selected; a full list of the AGEhIV Cohort Study group contributors can be found in Supplementary Note 1. We acknowledge and thank the NIAID Microbiome Program Sequencing Platform for performing DNA extraction, 16S rRNA sequencing, and the Metagenote team for data upload. The SomaScan Assay HTS system at the Center for Human Immunology is supported by NIH intramural funding (AI001226-01). B.S. is a former SomaLogic, Inc. (Boulder, CO, USA) employee and a company shareholder. Additional unrestricted scientific grants were received from Gilead Sciences, ViiV Healthcare, Janssen Pharmaceuticals N.V., and Merck & Co. None of these funding bodies had a role in the design or conduct of the study, the analysis or interpretation of the results, the writing of the report, or the decision to publish.

## Author contributions

I.S., O.S., P.R., N.K., M.S.v.d.L., J.B., J.A., and I.V.-C. designed the study. F.W.W., I.V.C., O.S., E.V., and N.K. participated in subject sample selection. I.V.-C. and O.S. performed microbiota analyses and wrote the manuscript with contributions from all authors. I.V.-C. and J.S. performed machine learning analyses. B.S. performed proteomic quantifications and data normalization, overseen by Y.B.

## Competing interests

PR reports independent scientific grant support from Gilead Sciences, Merck & Co, and ViiV Healthcare through his institution; scientific advisory board participation for Gilead Sciences, ViiV Healthcare, Merck & Co., and Teva Pharmaceutical Industries Ltd., for which his institution has received remuneration. The remaining authors declare no competing interests.
