## [Peer Review File · Nature Communications]

REVIEWERS' COMMENTS:

Reviewer #1 (Remarks to the Author):

I have reviewed the revised manuscript and the reviewer comments/responses.

This work contributes novel data to the field based on its relatively large sample size and relatively thorough careful selection of HIV negative controls with robust assessment of potentially confounding factors.

It is well written and clearly understandable.

The authors appear to have adequately addressed the reviewers comments with the exception of the discussion of ethnicity. While nation of birth is a helpful piece of data, it does not substitute for ethnicity and if ethnicity was not collected, it should be noted as a limitation.

Reviewer #2 (Remarks to the Author):

Re-Review of revised submission now to Nature Communications 2/2020

Gut dysbiosis is present in treated HIV infection independently of sexual practice and is linked to higher prevalence of non-communicable diseases

Overall remarks to the authors.

- Glad some of the comments from last review were of help. I've commented below on each point.
- These cross-sectional analyses (optimal design) were very helpful for the field; such well-powered investigations are crucial. The well-supported title that sexual practice (and not really gender) contributes to altered/dysbiosis advances information. Importantly, as good studies do, many points are raised for future investigation.

o GRANTED: The authors will likely feel they have sully addressed the following points related for sample # per subject in response #2 to this reviewer's comments from first submission. Fine. My comments remain below.

o There are many hypothesis-generating mechanisms proposed in the Discussion. The large n and the 3 comparison groups makes that possible. COMMENT: The Discussion posits many derivative hypotheses; these are based on a single baseline sample. To the extent there would have been mild/moderate/larger variability in several stool samples over a period of time, I'd imagine that factor would be carried forward (augmented?) in hypothesis testing based on this trial's findings. The authors are far better stats experts and will be able to address this question with the Editors.

o COMMENT: If valid or invalid, a clarification in the discussion up front would help, likely in the first Discussion paragraph. It may say: 'While some feel the lack of longitudinal and/or characterizations of these individuals stool microbiome variances over a month or so, the cross-over design as well as ...(included previously published JID findings) minimize this confounder.' (see next point).

o The results from one author's JID May 201 publication reports that human rectal microbiome samples collected over months show high concordance in phyla, taxa over several months. This supports the author's comments to Reviewer that "Furthermore, in our hands, longitudinal samples can be very reproducible by samples collected the same way and if no drastic diet or medication or health alterations have occurred." Sortino et al. JID May 2019.

☒ DIFFERENCES: Samples for this 'Gut dysbiosis...' manuscript were collected at home, refrigerated, stored and brought to clinic within 24 hours. In contrast, the JID paper microbiome subset samples were obtained fresh in clinic with a rectal swab.

☒ COMPARABILITY: The microbiome subset of the well-done Phase 1 trial cited as support for reasonably using a single stool sample in these cross-sectional analyses is based on data from 12 of 54 HIV+ subjects. Microbiome samples were collected (rectal swab/in clinic/by research team) 4 times over at least 8+ month study). Results show samples from the same subject clustered strongly (presuming this means baseline for Period 1 and baseline (Visit 5) for Period 2).

- Accurate that they emphasize association only, at this point. Good.
- Table 1 – Demographics. Significant effort went in to increasing descriptive characteristics of the study population. Additions include economic/education levels, some behaviors

(smoking/alcohol/amphetamine use. COMMENT: In a study focused on association of sexual practices with different microbiome ('stable') populations and those having associations with co-morbidities would reasonably want to have as detailed as possible descriptors of the 'sexual practices' involved.

o COMMENT: I'm sure the relevant subject details were not collected within the main parent study, limiting the author's inclusion. This caveat needs to be made a few times, especially give the large n of this study and possible interpretations/next studies. An inference would be that a single episode of RAI 5.5 months ago has a similar effect on the microbiome's changed diversity and the durability of that new microbiome profile to a subject having RAI several times a week/month – more so if with different partners. No way they can address these quite potent confounders in the definition of 'sexual practices'; therefore, this caveat needs inclusion. [For an example of the diversity of descriptors, see: Baggaley, R.F et al. Increases in HIV incidence following receptive anal intercourse among women: A systematic review and meta-analysis. AIDS Behav. 2019 Sep 4.].

- Supplementary Table #S11: extensive; layout more clearly shown, including comparisons (Binary, Shannon, Dysbiosis).
- Supplementary Figure #S1, S2: Both with improved clarity.
- Assay variability: My point in the earlier review was requesting information to give an indication of inherent variability in the assay(s) used (intra-assay; intra-subject; inter-subject). In most cases this information is acquired before use and did not mean to imply it should be derived from the trial data set. The advantage lies in further refinement of trial's power determinations as well as cumulative variabilities contributing to the final data's variances. COMMENT: if there are data quantifying the assay's inherent variability when the same sample is run twice through the complete (and fractional) assay steps (intra-assay variability) and ability to show reproducibility from two human samples acquired at the same time, same location (intra-subject variability), it would be helpful to include in Methods.

Reviewer #3 (Remarks to the Author):

Vujkovic-Cvijin et al examine HIV associated alterations in the stool microbiome, taking into account a number of demographic and behavioral factors in the analysis. Samples from the AGEHIV cohort based in Amsterdam are used for the analysis. The authors identify specific gut microbial changes associated with MSM behavior and further associate these changes with those who engage in receptive anal intercourse. They also identify an HIV-associated effect independent of sexual practices. In responses to prior reviewer comments they have added consideration of additional metadata as well as performed proteomic analysis in a subset of individuals. The proteomic analysis

identifies several inflammatory markers enriched in HIV infection. Overall, the manuscript is well written and the analyses are thorough and well presented. Although many of the higher level conclusions have been addressed by other reports, this manuscript help provides additional insight by carefully considering potential confounders, having greater details about sexual practices, utilizing a large cohort, and performing additional proteomic analysis. The authors have been responsive to prior reviewer comments, which I largely agree with, and I believe the revised manuscript would be of significant interest to the readership of Nature Communications.

REVIEWERS' COMMENTS:

We thank the reviewers for their comments.

Reviewer #1 (Remarks to the Author):

I have reviewed the revised manuscript and the reviewer comments/responses.

This work contributes novel data to the field based on its relatively large sample size and relatively thorough careful selection of HIV negative controls with robust assessment of potentially confounding factors.

It is well written and clearly understandable.

The authors appear to have adequately addressed the reviewers comments with the exception of the discussion of ethnicity. While nation of birth is a helpful piece of data, it does not substitute for ethnicity and if ethnicity was not collected, it should be noted as a limitation.

We have expanded discussion of this in the paragraph comprising caveats within the Discussion:

“Furthermore, while country of birth and immigration status were collected in our metadata, ethnicity information was not and may be an important potential confounding factor to collect in future studies.”

Reviewer #2 (Remarks to the Author):

Re-Review of revised submission now to Nature Communications 2/2020

Gut dysbiosis is present in treated HIV infection independently of sexual practice and is linked to higher prevalence of non-communicable diseases

Overall remarks to the authors.

- Glad some of the comments from last review were of help. I've commented below on each point.
- These cross-sectional analyses (optimal design) were very helpful for the field; such well-powered investigations are crucial. The well-supported title that sexual practice (and not really gender) contributes to altered/dysbiosis advances information. Importantly, as good studies do, many points are raised for future investigation.
 - o GRANTED: The authors will likely feel they have sully addressed the following points related for sample # per subject in response #2 to this reviewer's comments from first submission. Fine. My comments remain below.
 - o There are many hypothesis-generating mechanisms proposed in the

Discussion. The large n and the 3 comparison groups makes that possible.
COMMENT: The Discussion posits many derivative hypotheses; these are based on a single baseline sample. To the extent there would have been mild/moderate/larger variability in several stool samples over a period of time, I'd imagine that factor would be carried forward (augmented?) in hypothesis testing based on this trial's findings. The authors are far better stats experts and will be able to address this question with the Editors.

Indeed, if we had performed longitudinal studies in these subjects, we would have been able to propose hypotheses relating to temporal variability. Given that our study was cross-sectional in nature, we do not feel confident in making many statements or conclusions based on microbiota stability, but have expanded discussion on the potential utility of longitudinal studies in the Discussion:

“Longitudinal studies may also help disambiguate microbiota signals from confounding factors, though some lifestyle factors may change with diagnosis and should likely be continuously monitored. Relative longitudinal stability of the gut microbiota also was not examined in the present study, though prior studies have noted limited intra-individual dispersion among PWH^{67,68}. However, we cannot exclude that HIV infection induces microbiota instability in humans, which was not able to be captured in the present cross-sectional study design.”

We have also substantially expanded discussion of future longitudinal studies as follows:

“Non-human primate studies offer an attractive path for longitudinal analysis, though challenges in translating SIV microbiota studies exist in that SIV infection does not durably alter the macaque gut microbiota through the early phases of chronic infection⁷⁰⁻⁷². However, wild chimpanzee studies have reported microbiota instability in advanced SIV infection^{73,74}, and experimentally-infected rhesus macaques similarly exhibit microbiota alterations during advanced disease characterized by low CD4 counts⁷⁵ not unlike those described herein (i.e. *Enterobacteriaceae* bloom). Given that nadir CD4 count correlates in our study with extent of dysbiosis, it is possible that the inflammation and/or immune destruction occurring in advanced HIV and SIV disease causes microbiota disruptions that persist even after CD4 restoration via treatment, and that this altered microbiota contributes to elevated comorbidity burden in PWH. Future prospective longitudinal studies in humans and non-human primates are warranted to further dissect the impact of the observed specific shifts in the gut microbiota on associations with chronic inflammation and comorbidity prevalence in HIV as reported herein.

o COMMENT: If valid or invalid, a clarification in the discussion up front would help, likely in the first Discussion paragraph. It may say: ‘While some feel the lack of longitudinal and/or characterizations of these individuals stool microbiome

variances over a month or so, the cross-over design as well as ... (included previously published JID findings) minimize this confounder.' (see next point).

o The results from one author's JID May 2011 publication reports that human rectal microbiome samples collected over months show high concordance in phyla, taxa over several months. This supports the author's comments to Reviewer that "Furthermore, in our hands, longitudinal samples can be very reproducible by samples collected the same way and if no drastic diet or medication or health alterations have occurred." Sortino et al. JID May 2019.

♣ DIFFERENCES: Samples for this 'Gut dysbiosis...' manuscript were collected at home, refrigerated, stored and brought to clinic within 24 hours. In contrast, the JID paper microbiome subset samples were obtained fresh in clinic with a rectal swab.

Indeed. Given that all samples within each study were collected in a consistent manner, it is unlikely that comparisons of samples within each study would bear bias related to study-intrinsic collections. Furthermore, numerous studies have shown limited to undetectable microbiota shifts when storing samples at 4C for 24 hours (Choo JM et al. Sci Rep 2015, Tedjo et al. PLoS One 2015).

♣ COMPARABILITY: The microbiome subset of the well-done Phase 1 trial cited as support for reasonably using a single stool sample in these cross-sectional analyses is based on data from 12 of 54 HIV+ subjects. Microbiome samples were collected (rectal swab/in clinic/by research team) 4 times over at least 8+ month study). Results show samples from the same subject clustered strongly (presuming this means baseline for Period 1 and baseline (Visit 5) for Period 2).

We agree with this statement.

- Accurate that they emphasize association only, at this point. Good.
- Table 1 – Demographics. Significant effort went in to increasing descriptive characteristics of the study population. Additions include economic/education levels, some behaviors (smoking/alcohol/amphetamine use. COMMENT: In a study focused on association of sexual practices with different microbiome ('stable') populations and those having associations with co-morbidities would reasonably want to have as detailed as possible descriptors of the 'sexual practices' involved.

We have added insertive anal intercourse to our cohort characteristics (Supplementary Table 1), and have further discussed the possible implications of sexual practice in the caveats section of the Discussion: "Furthermore, given the potent impact of sexual preference on the gut microbiota, it is conceivable that sexual activities not captured in the present dataset may impact the microbiota signatures described herein, warranting further study."

o COMMENT: I'm sure the relevant subject details were not collected within the main parent study, limiting the author's inclusion. This caveat needs to be made a few times, especially give the large n of this study and possible interpretations/next studies. An inference would be that a single episode of RAI 5.5 months ago has a similar effect on the microbiome's changed diversity and the durability of that new microbiome profile to a subject having RAI several times a week/month – more so if with different partners. No way they can address these quite potent confounders in the definition of 'sexual practices'; therefore, this caveat needs inclusion. [For an example of the diversity of descriptors, see: Baggaley, R.F et al. Increases in HIV incidence following receptive anal intercourse among women: A systematic review and meta-analysis. AIDS Behav. 2019 Sep 4.].

We agree with this point and for this reason have added discussion of this point described in the response above.

- Supplementary Table #S11: extensive; layout more clearly shown, including comparisons (Binary, Shannon, Dysbiosis).
- Supplementary Figure #S1, S2: Both with improved clarity.
- Assay variability: My point in the earlier review was requesting information to give an indication of inherent variability in the assay(s) used (intra-assay; intra-subject; inter-subject). In most cases this information is acquired before use and did not mean to imply it should be derived from the trial data set. The advantage lies in further refinement of trial's power determinations as well as cumulative variabilities contributing to the final data's variances. COMMENT: if there are data quantifying the assay's inherent variability when the same sample is run twice through the complete (and fractional) assay steps (intra-assay variability) and ability to show reproducibility from two human samples acquired at the same time, same location (intra-subject variability), it would be helpful to include in Methods.

For all components of our pipeline we have worked to utilize standard approaches that are well-established in the field of the gut microbiome and have been tested for reproducibility. We have utilized standard primers employed by worldwide microbiome characterization initiatives such as the Earth Microbiome Project and Integrative Human Microbiome Project (iHMP2), commonly used commercial kits (PowerMicrobiome DNA/RNA EP Kit), and well-validated analysis pipelines (dada2, over 2,000 citations since publication in 2016). Furthermore, we believe intra-assay variability is unlikely to produce the associations we report; ecological community comparisons used herein (PERMANOVA) are agnostic to total intra-sample variability and only consider significant microbial trends that are consistent across groups of samples. The specific ASV-level group comparisons we employed also preferentially identify robust trends existing across samples within groups, which are themselves consistent

with prior literature. Finally, our samples were processed in 96-well plates; so as to eliminate the effects of inter-batch differences, PWH and seronegative controls were evenly balanced per 96-well plate, as were the three subject groups of MSM, MSW, and female (which can be verified in metadata of SRA database entry SRP229524, column 'Library Name'). There were no ecological community-based differences between the plates detected, as seen in Supplementary Table 3 (category 'DNA extraction batch'), suggesting a lack of systematic inter-assay variability. We feel that adding individual subjects as biological replicates in our cross-sectional study confers strengths that are missing from profiling the same number of samples with multiple samples from fewer individuals or the same samples numerous times. Specifically, generalizability of our findings is more likely given the large number of individual subjects as compared to a more limited number of subjects with numerous samplings. Thus, it can be said that the cross-sectional study design forms a credible first step in addressing confounding factors in HIV microbiota studies, the principal goal of our work. We believe a concrete future direction that is likely to yield important insights lies in longitudinal studies, hence our focus on it in the Discussion, which we have expanded as noted above. Nevertheless, with regards to stability within our sequencing data, when re-sampling 10,000 reads twice from the same sample and computing a Principal Coordinates Analysis on a Canberra distance matrix of the replicates, we find tight clustering by sample that overcomes inter-sample differences:

Reviewer #3 (Remarks to the Author):

Vujkovic-Cvijin et al examine HIV associated alterations in the stool microbiome, taking into account a number of demographic and behavioral factors in the analysis. Samples from the AGEHIV cohort based in Amsterdam are used for the analysis. The authors identify specific gut microbial changes associated with

MSM behavior and further associate these changes with those who engage in receptive anal intercourse. They also identify an HIV-associated effect independent of sexual practices. In responses to prior reviewer comments they have added consideration of additional metadata as well as performed proteomic analysis in a subset of individuals. The proteomic analysis identifies several inflammatory markers enriched in HIV infection. Overall, the manuscript is well written and the analyses are thorough and well presented. Although many of the higher level conclusions have been addressed by other reports, this manuscript help provides additional insight by carefully considering potential confounders, having greater details about sexual practices, utilizing a large cohort, and performing additional proteomic analysis. The authors have been responsive to prior reviewer comments, which I largely agree with, and I believe the revised manuscript would be of significant interest to the readership of Nature Communications.

Thank you!